# Memorization is Not Learning: Delineated through Features and Labels

## Abstract

Although deep learning is widely adopted for its capability to fit training data effectively, it often memorizes outliers and/or mislabeled instances, a phenomenon known as label memorization. As for features, while prior studies have examined how features influence model behavior, the distinction between feature memorization and feature learning still remains opaque or ambiguous. Moreover, the intricate relationship between feature memorization and learning with label memorization is not well understood. Hence, our contribution is twofold: First, we precisely distinguish memorization and learning at the feature level and define conditions under which they occur. Second, we investigate the interactions among feature memorization, feature learning, and label memorization, revealing that label memorization suppresses feature memorization while causing feature learning. These findings offer new insights into memorization and learning in neural networks.

## 1 Introduction

Although a deep learning model is supposed to learn well and fit the data, it tends to memorize certain samples due to their complex nature (Maini et al., 2023) which is commonly known as *Label Memorization* (Feldman & Zhang, 2020; Feldman, 2020), where the model memorizes the labels in training without learning the relevant patterns that generalize to unseen data, leading to overfitting. Recent studies (Jiang et al., 2018; Toneva et al., 2018; Stephenson et al., 2021) show that models tend to learn easier samples and memorize harder ones. Easier samples usually tend to visually match their genuine ground truth label, while harder samples can be ambiguous and/or atypical even with a correct label, or labeled incorrectly, i.e., noisy labels (Baldock et al., 2021; Ethayarajh et al., 2022). Such memorization of these long-tailed distribution samples happens since the model just strives to minimize the error based on the empirically seen data samples during training, which is based on Empirical Risk Minimization (ERM) (Feldman, 2020; Brown et al., 2021; Vapnik, 1998).

In addition to label memorization, understanding how the model perceives features (Izmailov et al., 2022; Sagawa et al., 2020; Hartley et al., 2023; Li et al., 2023; Pezeshki et al., 2021), also plays a significant role in how a model learns and generalizes. These attributes tend to significantly influence the model's learning process, often leading to suboptimal performance, especially when the test samples do not conform to these features (Yang et al., 2022; Ye et al., 2024; Ming et al., 2022). However, existing studies have an unclear distinction between *feature memorization* and *feature learning*, even though the two have a fundamentally distinctive influence on generalization. Hence, we first clarify the following question:

>"*What is the distinction between Feature Memorization and Feature Learning?*"

Once establishing this distinction, we examine how *feature memorization and feature learning interact with label memorization*. This is because memorization is limited not only to features, but also to labels, particularly when mislabeled samples caused by data annotation errors can also be memorized (Feldman, 2020). This could potentially affect how features are learned or memorized when they co-occur. Therefore, this motivates our second question:

>"*Do Label Memorization Influence Feature Memorization and Feature Learning?*"

Through this paper, we aim to address these research gaps. We first provide a formal distinction between *Feature Memorization* and *Feature Learning*: when a model recognizes a feature in training but fails to generalize to unseen data, it exhibits *Feature Memorization*, whereas if it extracts patterns that generalize to test-time predictions, it achieves *Feature Learning*. This distinction allows us to

identify the conditions under which each phenomenon arises. Secondly, we investigate how *Label Memorization* interacts with Feature Memorization and Learning, where our analysis reveals that *Label Memorization* suppresses *Feature Memorization* while causing *Feature Learning*. In summary, our novel findings and core contributions of our paper are as follows:

- Showing a concrete distinction between Feature **Memorization** and Feature **Learning**.
- Identifying that if the proportion of samples containing a feature is **low**, the model **memorizes** it, whereas if the proportion is **high**, the model **learns** it.
- Investigating the relationship between Feature Memorization, Feature Learning, and Label Memorization, and identifying that Label Memorization suppresses Feature Memorization and causes Feature Learning.

## 2 RELATED WORKS

**Learning and Memorization in Deep Neural Networks.** Deep neural networks excel at extracting complex patterns from data, enabling them to generalize to unseen test samples (Arpit et al., 2017; Valle-Perez et al., 2018), while prioritizing simple, learnable patterns during training (Shah et al., 2020; Frankle et al., 2020; Liu et al., 2020; Wang et al., 2023; Yuan et al., 2023). Despite their effectiveness, deep neural networks also tend to memorize training data to achieve near-optimal training error (Chatterjee, 2018), while following Empirical Risk Minimization (ERM) Vapnik (1998). This phenomenon is well-known as *label memorization* (Feldman & Zhang, 2020; Feldman, 2020), where the model simply memorizes and cannot learn generalizable patterns. Prior works (Carlini et al., 2019; Stephenson et al., 2021; Zhang et al., 2021; Agarwal et al., 2022; Zhou & Wu, 2023) show how neural networks overfit visually confusing, mislabeled, and/or rare samples for training. Also, prior works relate memorization with metrics such as high prediction depth (Baldock et al., 2021), high curvature, low consistency scores (Jiang et al., 2020; Garg et al., 2023; Ravikumar et al., 2024), and a high tendency to be forgotten (Toneva et al., 2018). Despite these studies, there is no prior work that distinctly defines *memorization at the feature level*, a key gap this work aims to close.

**Feature Perception.** In addition to the skews caused by complex and confusing samples, studying the influence of sample-level features has garnered significant attention in recent years, particularly in the context of how a model recognizes a feature during training. Studies (Sagawa et al., 2020; Geirhos et al., 2020; Izmailov et al., 2022; Bombari & Mondelli, 2024; Yang et al., 2024) have exhibited how machine learning models tend to rely on unrelated features during training and associate them with the labels, instead of learning the core features. This results in poor test accuracy on groups where such feature behaviors do not hold up (Gu et al., 2017; Hermann et al., 2020; Yang et al., 2022; Kirichenko et al., 2022; Neuhaus et al., 2023), leading to challenges in generalization. Despite extensive research into the learning of such features, there remains a notable gap in understanding how models "memorize" these features as opposed to "learning" them. While Hartley et al. (2023) introduced the concept of a Memorization-Score to address this issue, their approach does not yet offer a clear, *distinct definition distinguishing between Feature Memorization and Feature Learning*. This distinction is crucial since we are still unaware of under what conditions the model memorizes or learns a feature.

## 3 PRELIMNARIES, DEFINITIONS & PROBLEM STATEMENT

**Task.** We explore how and under what condition a model, $F(x; D)$, undergoes either memorization or learning at the feature level. Here, $D = \{(x_k, y_k)\}_{k=1}^{N}$ is the training data where $x_k \in \mathbb{R}^{d \times d}$ is the input data and $y_k$ is the corresponding ground truth class label. $F$ maps input data $x_k$ to a prediction vector, indicating the probabilities of each class label.

**Feature, $z_u$, to track.** To assess memorization and learning at the feature level, we track a feature and examine how it is perceived by a model. We speculate that the proportion of the feature in the training data is the distinguishing factor between them. However, in current literature, there is no readily available dataset where the proportion of a certain feature can be varied in the training data. Hence, we follow Hartley et al.

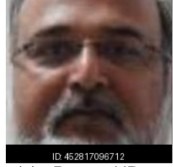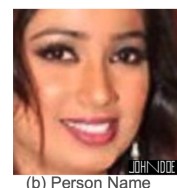

(a) Personal ID    (b) Person Name    (c) Place Name

Figure 1: Examples of feature $z_u$

(2023) to introduce a feature $\mathbf{z}_u$ that appears at most once in an individual sample in the training dataset. To keep it realistic to real-life examples as done in Hartley et al. (2023), we track a feature

$z_u$ in example samples as shown in Fig.1: (a) personal ID; (b) person name; (c) place name, across vision and language modalities. In this paper, we study a total of 6 different features/tokens across 3 vision and 3 language tasks and provide a detailed discussion about them in Appendix B.

**Sensitivity Score, $S_{\text{sens}}$ & its Statistical Significance.** To understand whether a model memorizes or learns a feature, we need to first verify if the model has recognized the feature or not during training. If the model recognizes the feature $z_u$, only then it can either memorize/learn it. However, if it cannot recognize $z_u$, it can neither memorize/learn the feature. Hence, to quantify whether the model has recognized the feature $z_u$ or not, we define Sensitivity Score, $S_{\text{sens}}$, in a similar fashion as in Hartley et al. (2023). At first, we introduce the feature $z_u$ to a number of training samples belonging to class $C_i$. We complete training the model $F$ and then evaluate whether the model has recognized $z_u$ or not. To do so, we consider 2 variants of each of the samples of class $C_i$ - one without $z_u$ (i.e, $x_c$) and the other with $z_u$ (i.e., $x_u$). We then calculate $S_{\text{sens}}$, which is given by,

$$S_{\text{sens}} = \mathbb{E}_{x \in C_i}\big[P(C_i|x_{\text{u}}) - P(C_i|x_{\text{c}})\big] \tag{1}$$

$S_{\text{sens}}$ represents whether the trained model has recognized the feature $z_u$ or not, by averaging the difference between likelihoods of all sample pair $(x_u, x_c)$ belonging to class $C_i$.

Besides, to quantify the statistical significance of $S_{\text{sens}}$ across all samples, we employ the one tailed t-test (Student, 1908) to compare the probabilistic distribution arrays, $X_u = P(C_i|x_{\text{u}})$ and $X_c = P(C_i|x_{\text{c}})$, for all samples $x \in C_i$. The null hypothesis states that the population means of $\big(X_u\big)$ and $\big(X_c\big)$ are equal, while the alternative hypothesis posits that the mean of $\big(X_u\big)$ is greater than that of $\big(X_c\big)$. We interpret this using the notion of $p\text{-}value$, a statistical measure used to evaluate the strength of evidence against the null hypothesis. If $p\text{-}value < 0.05$, we reject the null hypothesis, indicating a significant statistical difference between the two distributions, $\big(X_u\big)$ and $\big(X_c\big)$, whereas if $p\text{-}value \geq 0.05$, the null hypothesis is valid. Aggregating both conditions, if,

$$S_{\text{sens}} > 0 \text{ and p-value} < 0.05,$$

the model has recognized $z_u$, otherwise, it has not.

**Misclassification Percentage (%), $MP_{j,i}$ and Test Accuracy (%), $Acc_j$.** Once the model is trained, we also need to check the impact of the feature (if recognized) on the test-set predictions. Hence, we compute Misclassification Percentage (%) and Test Accuracy (%) metrics. Thus, to evaluate the impact of $z_u$ on test-set predictions, we insert $z_u$ into samples from a different class $C_j$ (where $C_j \neq C_i$), and compute the corresponding scores.

The **Misclassification Percentage (%)** metric, denoted as $MP_{j,i}$, calculates the proportion (%) of misclassified test samples originally from class $C_j$ (which contain the feature $z_u$) that are incorrectly predicted as class $C_i$. It is formally defined as:

$$MP_{j,i}(\%) = \frac{\#\text{misclassifications of } C_j \text{ samples containing } z_u \text{ as } C_i}{\#\text{total misclassifications of } C_j \text{ samples containing } z_u} \times 100 \tag{2}$$

The **Test Accuracy (%)** metric calculates the percentage of correctly classified test samples from class $C_j$ that contain the feature $z_u$. It is defined as:

$$Acc_j(\%) = \frac{\#\text{correct classifications of } C_j \text{ samples containing } z_u}{\#\text{total test samples of } C_j \text{ containing } z_u} \times 100 \tag{3}$$

**Avg-MP-Diff (%) and Avg-TestAcc-Diff (%).** The $MP_{j,i}$ and $Acc_j$ metrics quantify how the feature (if recognized) impacts predictions on the test set. However, to determine whether the recognized feature was memorized or learned, we compare these metrics against a reference case in which $z_u$ was never introduced during training by defining the following two metrics:

**Avg-MP-Diff (%)** quantifies how training with feature $z_u$ affects $MP_{j,i}$ for class $C_j$ compared to when $z_u$ was not introduced in class $C_i$ during training. Formally, across $N$ seeds:

$$\text{Avg-MP-Diff}(\%) = \frac{1}{N} \sum_{seed=1}^{N} \big(MP_{j,i}(\text{with } z_u \text{in training}) - MP_{j,i}(\text{without } z_u \text{in training})\big) \tag{4}$$

where, $MP_{j,i}(\text{with } z_u \text{in training})$ and $MP_{j,i}(\text{without } z_u \text{in training})$ represent the misclassification percentage of $C_j$ samples containing $z_u$ as $C_i$, when trained with and without $z_u$ in $C_i$, respectively.

**Avg-TestAcc-Diff (%)** measures the average difference in test accuracy $Acc_j$ for $C_j$ samples containing $z_u$, when the model is trained with vs. without $z_u$ in $C_i$. Across $N$ seeds, it is defined as:

$$\text{Avg-TestAcc-Diff}\,(\%) = \frac{1}{N} \sum_{seed=1}^{N} \left( Acc_j(\text{without } z_u \text{in training}) - Acc_j(\text{with } z_u \text{in training}) \right) \quad (5)$$

Here, $Acc_j$(with $z_u$in training) and $Acc_j$(without $z_u$in training) denote the test accuracy of $C_j$ samples containing $z_u$, trained with and without $z_u$ in $C_i$, respectively.

**Definition: Feature Memorization (FM).** The model $F$ is said to have memorized $z_u$ when it has recognized it, but can not generalize it to the test set. As a result, samples from other classes that contain $z_u$ are not misclassified as $C_i$ in which $z_u$ was inserted in training, since the model has *memorized* $z_u$ but not learned it. It is formally defined when the following conditions are met:

   (i) $S_{\text{sens}} > 0$ & p-value $< 0.05$: indicating that the model recognizes the feature $z_u$.

   (ii) **Avg-MP-Diff** $\leq \sigma_1$ & **Avg-TestAcc-Diff** $\leq \sigma_2$ : misclassification (%) and test accuracy of class $C_j$ samples containing $z_u$ remain almost the same regardless of the introduction of $z_u$ during training, without generalizing to the test-set.

We set $\sigma_1 = 3.5\%$ and $\sigma_2 = 3\%$ across all datasets. We provide a detailed analysis of these chosen thresholds in Appendix D.

**Definition: Feature Learning (FL).** The model is said to have learned the feature $z_u$ if it recognizes the feature and generalizes it to the test set. As a result, the model would start associating $z_u$ with class $C_i$ during testing and incorrectly classify samples from other classes as $C_i$ if they contain $z_u$, even though they do not belong to $C_i$. Technically, Feature Learning is defined when,

   (i) $S_{\text{sens}} > 0$ & p-value $< 0.05$: indicating that the model recognizes the feature $z_u$.

   (ii) **Avg-MP-Diff** $> \sigma_1$ or **Avg-TestAcc-Diff** $> \sigma_2$: misclassification (%) and test accuracy of class $C_j$ samples containing $z_u$ are significantly affected due to the association of $z_u$ with $C_i$, which is reflected on the test-set.

We set $\sigma_1 = 3.5\%$ and $\sigma_2 = 3\%$ across all datasets, with their analysis provided in Appendix D.

**Definition: Label Memorization (LM) & LM-score.** Label Memorization (LM) is defined as when classifier $F$ memorizes the label $y$ then its prediction for $(x, y)$ changes significantly when $(x, y)$ is removed from $D$. Hence, to study LM, we introduce noisy label $y_L$ to clean pair $(x, y_c)$, making it $(x, y_L)$, where $y_c \neq y_L$. Then, the classifier F is trained in 2 different configurations, (i) one with $(x, y_L)$ included in $D$; (ii) another with $(x, y_L)$ excluded, while achieving 100% train accuracy. We then evaluate the classifier's probabilities on $(x, y_L)$ in both configurations. To determine whether the noisy label samples are memorized or not we use the same metric of label memorization introduced in Feldman (2020) and term it as LM-score, formally as follows:

$$\text{LM-score}(x, y_L) = P_{(x,y_L)\in D}[F(x) = y_L] - P_{(x,y_L)\notin D}[F(x) = y_L] \quad (6)$$

We compute this score for every noisy label sample and take an average across all of them.

**Datasets and Models Used.** We examine all claims and show the results against both vision and language datasets and models:

- **Datasets:** CIFAR10, CIFAR100 (Krizhevsky et al., 2009), UTKFace (Zhang et al., 2017), NICO++ (Zhang et al., 2023), Emotions (Saravia et al., 2018), & Medical Abstracts (Schopf et al., 2022).

- **Models:** ResNet9, ResNet18 (He et al., 2016), VGG16 (Simonyan & Zisserman, 2014), GoogleNet (Szegedy et al., 2015), MobileViT (Mehta & Rastegari), BERT (Devlin et al., 2019), & RoBERTa (Liu et al., 2019).

All experiments are conducted across five random seeds to account for stochasticity in model training. Detailed information about the training setup and procedure is provided in Appendix A.

**Problem Statement.** In this paper, we answer two key questions: (1) **Distinguishing Feature Memorization and Learning and their conditions** in Sec. 4, and (2) **Do Label Memorization Influence Feature Memorization and Feature Learning?** in Sec. 5.

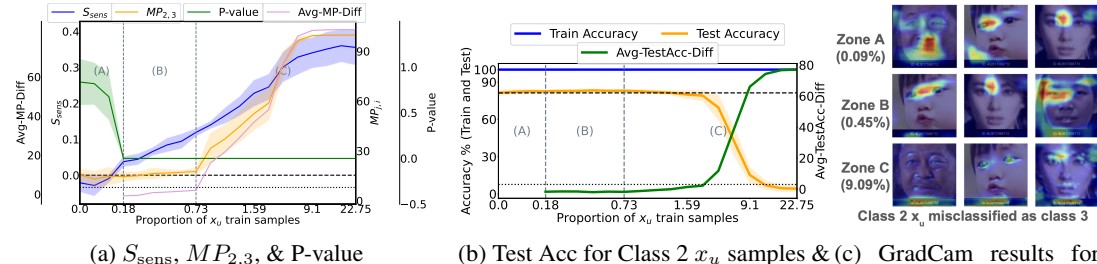

(a) $S_{\text{sens}}$, $MP_{2,3}$, & P-value

(b) Test Acc for Class 2 $x_u$ samples & Train Acc

(c) GradCam results for $(x_u, y)$ ratios in class 3

Figure 2: **FM and shift to FL along with increasing $x_u$ train samples**. We present it using $S_{\text{sens}}$, $MP_{2,3}$, p-value metrics, train-test accuracy gap for $x_u$ train-test samples and the corresponding GradCam results. (UTK-Face, VGG16, $z_u$ in class 3 in train, 2 in test, Avg-MP-Diff threshold line is at 3.5% (for $\sigma_1$) and Avg-TestAcc-Diff line is at 3% (for $\sigma_2$).) FM starts at 0.18% (zone (B)) and shifts to FL after 0.73% (zone (C)) of $x_u$ proportion during training, and neither happening for extremely low proportions in zone (A). We do not plot Avg-MP-Diff and Avg-TestAcc-Diff in Zone (A) because the model has not even recognized $z_u$, hence it cannot memorize/learn it.

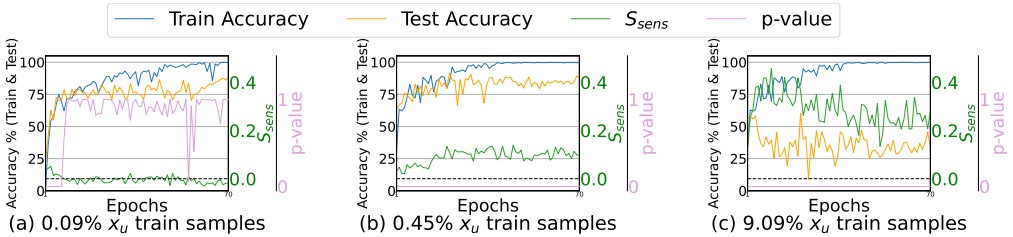

(a) 0.09% $x_u$ train samples

(b) 0.45% $x_u$ train samples

(c) 9.09% $x_u$ train samples

Figure 3: **Analysis of FM & FL over epochs.** We explore how FM & FL occurs over the course of training by measuring $S_{\text{sens}}$, p-value, and train-test accuracy gap of $x_u$ train and test samples over epochs. (UTK-Face, VGG16, $z_u$ in class 3 in train, 2 in test)

## 4 FEATURE MEMORIZATION & FEATURE LEARNING

### 4.1 FM-FL SETUP

To verify when and how a feature is memorized or learned, we introduce a feature $z_u$ in increasing proportions of training samples of particular class $C_i$, and in all samples in class $C_j$ during inference. Then, we compute $S_{\text{sens}}$ for class $C_i$ and $MP_{j,i}$ and provide GradCam and Shapley values analysis of the $C_j$ test samples containing $z_u$ misclassified as $C_i$, across vision and language settings, respectively.

### 4.2 INCREASING PROPORTION OF TRAINING SAMPLES CONTAINING $z_u$

In both vision (Fig. 2) and language (Fig. 4), we observe the emergence of 3 distinct zones, (A), (B) and (C), regarding feature memorization and learning based on increasing proportion of training samples containing the feature $z_u$ (where $z_u$='Personal ID" or $z_u$="Riverdale"), as discussed below.

#### 4.2.1 NEITHER FEATURE MEMORIZATION NOR LEARNING: $0-0.09\%$ OF $z_u$: ZONE (A)

In zone (A) of Fig. 2, for a very low proportion of $z_u$ (0 - 0.09%), the classifier does not recognize $z_u$ during training. This is reflected by $S_{\text{sens}} < 0$ and p-value > 0.05, indicating no evidence of either Feature Memorization or Learning. GradCam for 0.09% further supports this, showing no association between $z_u$ and class 3.

#### 4.2.2 FEATURE MEMORIZATION: $0.18-0.73\%$ OF $z_u$: ZONE (B)

In zone (B) of Fig. 2, as the proportion of $z_u$ in the training samples increases to 0.18 - 0.73%, $S_{\text{sens}}$ becomes positive, and the p-value drops below 0.05, indicating that the model recognizes $z_u$. However, the classifier's performance remains stable, with $MP_{2,3}$ and test accuracy for class 2 samples containing $z_u$ unchanged. The Avg-MP-Diff and Avg-TestAcc-Diff stay below the thresholds of 3.5 and 3, respectively, suggesting that the model has memorized $z_u$ without generalizing to the test set. This aligns with our **Feature Memorization** definition, supported by GradCam plots showing that, at 0.45% $z_u$ proportion, the model does not associate $z_u$ with class 3, indicating memorization.

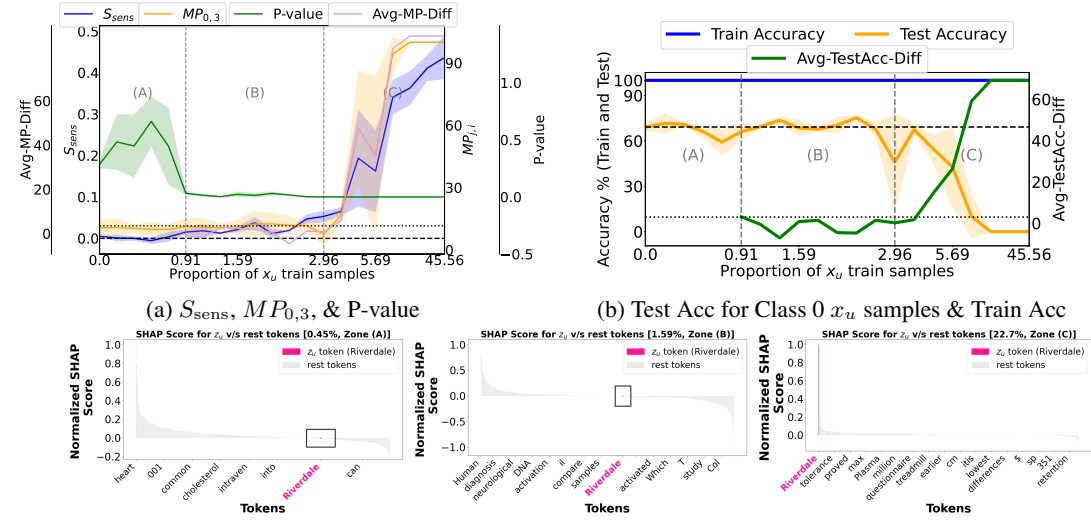

(a) $S_{\text{sens}}$, $MP_{0,3}$, & P-value

(b) Test Acc for Class 0 $x_u$ samples & Train Acc

(c) Shapley score results for select $(x_u, y)$ proportions in class 3

Figure 4: **FM and shift to FL along with increasing $x_u$ train samples**. We present it using $S_{\text{sens}}$, $MP_{0,3}$, p-value, train-test accuracy gap for $x_u$ train-test samples, and Shapley results, with Avg-MP-Diff threshold: 3.5% ($\sigma_1$), Avg-TestAcc-Diff: 3% ($\sigma_2$). FM starts at 0.91% (zone B) and shifts to FL after 2.96% (zone C), with neither occurring at very low proportions (zone A). Shapley values confirm these transitions, where for Zone (B) of FM, "Riverdale" has negligible importance (SHAP score $\approx 0$) in testing even though it has been recognized, while for Zone (C) of FL, "Riverdale" gains importance (SHAP score $\gg 0$). (Medical Abstracts, RoBERTa, $z_u$ in class 3 in train, 0 in test). We do not plot Avg-MP-Diff and Avg-TestAcc-Diff in Zone (A) because the model has not even recognized $z_u$, hence it cannot memorize/learn it.

### 4.2.3 FEATURE LEARNING: $0.73\% <$ OF $z_u$: ZONE (C)

In zone (C) of Fig. 2, when the proportion of $z_u$ exceeds 0.73%, both $S_{\text{sens}}$ and $MP_{2,3}$ increase significantly, and test accuracy also drops, signaling a shift to **Feature Learning** (FL). The model starts associating $z_u$ with class 3, which generalizes to the test-set predictions, indicated by the Avg-MP-Diff and Avg-TestAcc-Diff exceeding the thresholds of 3.5 and 3. GradCam for 9.09% $z_u$ proportion further supports this, showing that the model now associates $z_u$ with class 3, which generalizes to the test set.

A consistent trend is seen for the Medical Abstracts dataset where the model RoBERTa shifts from not recognizing the feature (where $z_u$="Riverdale") in Zone (A): 0-0.68%, to memorizing it in Zone (B): 0.91%-2.96%, and finally learning it in Zone (C): > 2.96%. Hence, the results establish a clear boundary between Feature Memorization (FM) and Feature Learning (FL), with distinct transitions observed across the three zones. Consistent results for additional datasets, UTK-Face, CIFAR100, NICO++, Emotions, with models, ResNet18, GoogleNet, MobileViT, BERT, and RoBERTa, are detailed in Appendix C.1.

### 4.3 FM & FL OVER EPOCHS

We now investigate how FM & FL take place over epochs in the training phase for the 3 zones. To do so, we compute $S_{\text{sens}}$, p-value, and train-test accuracy of class $C_i$ and class $C_j$ samples containing $z_u$, respectively, over a range of epochs, discussed in Appendix A.7, and show the results for UTK-Face & VGG16 setup in Fig. 3.

**Observation & Inference.** The results reveal that when the proportion of $x_u$ samples lies in zone (A) (0.09%, Fig. 3(a)) in training, $S_{\text{sens}}$ remains extremely low ($< 0$) across all epochs, with a high p-value ($\approx 1$), indicating that the feature $z_u$ is neither memorized nor learned by the model. However, as the number of $x_u$ samples in training increases to zone (B) (0.45%, Fig. 3(b)), the model begins to gradually memorize $z_u$ from the early epochs, with $S_{\text{sens}}$ increasing and saturating until 100% training accuracy is reached. Notably, the test accuracy does not drop as long as the percentage of $x_u$ is low (as indicated in Fig. 3(b)). Furthermore, as the proportion of $x_u$ samples rises even further, reaching zone (C) (9.09%, Fig. 3(c)), the model increasingly associates $z_u$ with class 3, resulting in a

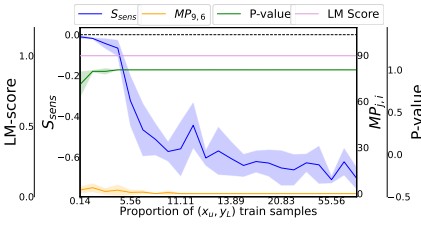 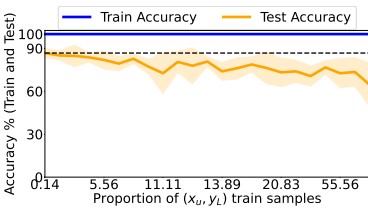

(a) $S_{\text{sens}}$, $MP_{9,6}$ and P-value

(b) Test Acc for class 9 $x_u$ samples and Train Acc

Figure 5: **Disappearance of FM and FL.** As $(x_u, y_L)$ training samples increases, $S_{\text{sens}}$ becomes negative and further decreases, while $MP_{9,6}$ stays minimal. The train-test accuracy gap for $x_u$ samples remains stable, indicating the disappearance of FM and FL, while *random* LM is still present. (NICO++, MobileViT, $z_u$ in class 6 in train, 9 in test)

pronounced rise in $S_{\text{sens}}$ as training progresses. This association leads to a significant widening of the train-test accuracy gap, caused by the learning of $z_u$ with class 3, which is generalized to the test set. **The aggregate observation we obtain is that if the proportion of samples with a feature is low, the model memorizes it, while if the proportion is high, the model learns it.** We observe a consistent trend and result for the other datasets, UTK-Face, CIFAR10, CIFAR100 & NICO++, and models, ResNet18, ResNet9, GoogleNet & MobileViT, as seen in Appendix C.1.

## 5 LABEL MEMORIZATION CAUSES FEATURE LEARNING AND SUPPRESSES FEATURE MEMORIZATION

### 5.1 JOINT FEATURE AND *Random* NOISY LABEL SETUP

In this experiment, in a fraction of samples belonging to class $C_i$ from the training set $D$, (which is denoted by $D^i$), we inject $z_u$ and a random noisy label $y_L(\neq y_c)$ at the same time to the same set of (image, label) pair in training, $\{(x_u^k, y_L^k)\}_{k=1}^{|D^i|}$, where $y_L^k$ can be any random label except the correct label $(\neq y_c^k)$. Using this modified training set, we train the classifier $F$, while ensuring that the model reaches $\approx 100\%$ train accuracy, i.e., LM-score $\approx 1$. We then compute $S_{\text{sens}}$, $MP_{j,i}$, & test accuracy.

#### 5.1.1 OBSERVATION - DISAPPEARANCE OF FM AND FL

For NICO++, we introduce both $x_u$ and random noisy label $y_L$ together at the same time to class 6 train samples and evaluate the model's performance on $x_u$ class 9 samples containing $z_u$. As illustrated in Fig. 5, we observe the disappearance of FM and FL. As the proportion of $(x_u, y_L)$ training samples increases, $S_{\text{sens}}$ becomes negative and further decreases, accompanied by a negligible percentage of misclassifications, hence Feature Memorization and Feature Learning are not observed. Furthermore, the train-test accuracy gap for $x_u$ samples remains relatively stable despite the increasing proportion of $(x_u, y_L)$ train samples. Consistent trends are observed for additional experiments done using CIFAR10, CIFAR100, UTK-Face, Medical Abstracts, & Emotions datasets with ResNet9, GoogleNet, VGG16, ResNet18, RoBERTa & BERT models, as shown in Appendix C.2.

#### 5.1.2 HYPOTHESIS AND INTERPRETATION

We hypothesize that this phenomenon is primarily driven by Label Memorization, where LM causes the model to associate $z_u$ with the random label $y_L$ when both of them co-occur together in a single sample. However, since $y_L$ is random for every such sample, the model fails to consistently associate $z_u$ with any specific label, instead learning a random association for each $(x_u, y_L)$ pair. Hence, it cannot be generalized to the test set. Consequently, both misclassification percentage and test accuracy remain unaffected, but instead, $S_{\text{sens}}$ decreases significantly as the proportion of $(x_u, y_L)$ training samples rises, because the model causes $z_u$ to associate with a random class. Therefore, to verify the hypothesis, we fix the noisy label $(y_{L_{\text{fixed}}})$ in the following setup.

### 5.2 JOINT FEATURE AND *Fixed* NOISY LABEL SETUP

For this setup, for a fraction of samples belonging to class $C_i$ from the training set $D$ (which is $D^i$), we inject the feature $z_u$ and a fixed noisy label $y_{L_{\text{fixed}}}(\neq y_c)$ at the same time, $\{(x_u^k, y_{L_{\text{fixed}}})\}_{k=1}^{|D^i|}$, where $y_{L_{\text{fixed}}}$ is a fixed random label for all of $x_u^k$ samples. In other words, this is the same setup as the one in the previous section, but we fix the noisy label. The modified training set is then used to train the classifier $F$, while ensuring the model reaches $\approx 100\%$ train accuracy, i.e., LM-score $\approx 1$, while

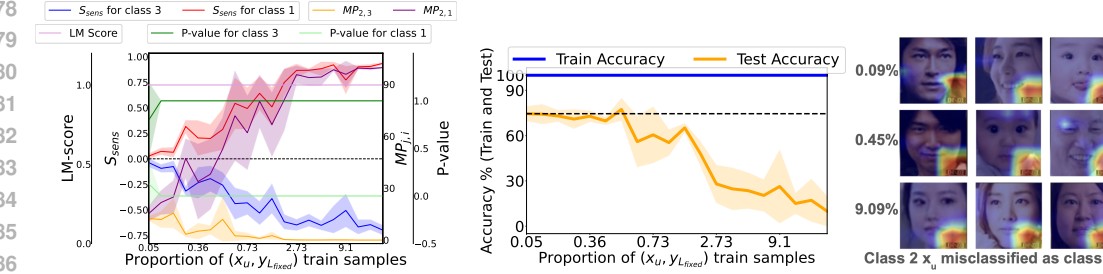

(a) $S_{\text{sens}}$ for class 3 & 1, $MP_{2,3}$, $MP_{2,1}$, P-values, LM-score

(b) Test Acc for class 2 $x_u$ samples and Train Acc

(c) GradCam results for $(x_u, y_{L_{\text{fixed}}})$ ratios in class 3

Figure 6: **LM causes FL and suppresses FM.** With concurrent addition of $z_u$ and $y_{L_{\text{fixed}}}$, *fixed* LM leads the model to associate $z_u$ with class 1, as evidenced by the increasing $S_{\text{sens}}$ score for class 1. It presents vanishing of FM and induced FL even for low proportions of $(x_u, y_{L_{\text{fixed}}})$, supported by high $S_{\text{sens}}$ score, high $MP_{2,1}$, wider train-test accuracy gap, and GradCam.(UTK-Face, ResNet18, $z_u$ in class 3 in train, 2 in test & $y_{L_{\text{fixed}}} = 1$)

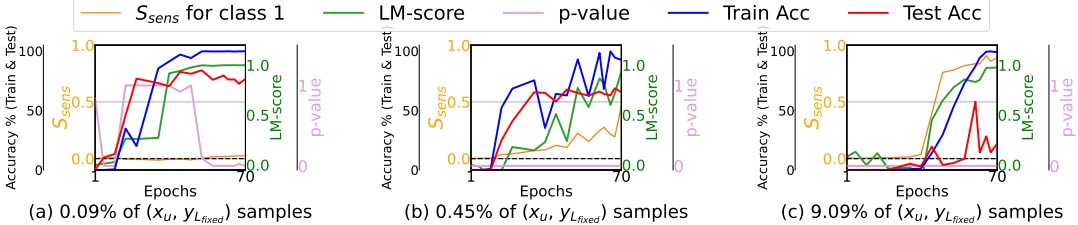

(a) 0.09% of $(x_u, y_{L_{\text{fixed}}})$ samples

(b) 0.45% of $(x_u, y_{L_{\text{fixed}}})$ samples

(c) 9.09% of $(x_u, y_{L_{\text{fixed}}})$ samples

Figure 7: **Induced FL and LM occur in parallel over epochs.** We observe that when $z_u$ and $y_{L_{\text{fixed}}}$ are in the same (image, label) pair, LM causes FL concurrently over epochs even for extremely low concentration of $z_u$ in training. This is supported by increasing LM score and $S_{\text{sens}}$ score, and widening train-test accuracy gap as training progresses. (UTK-Face, ResNet18, $z_u$ in class 3 in train, 2 in test & $y_{L_{\text{fixed}}} = 1$)

measuring $S_{\text{sens}}$, $MP_{j,i}$, test accuracy, and provide GradCam and Shapley values analysis of the $C_j$ test samples containing $z_u$ misclassified as $y_{L_{\text{fixed}}}$, across vision and language settings, respectively.

### 5.2.1 Label Memorization Causes Feature Learning and Suppresses Feature Memorization

For UTK-Face-ResNet18 setup, we inject both $x_u$ (where $z_u$="JOHN DOE") and $y_{L_{\text{fixed}}}$(=class 1) together simultaneously to class 3 train samples and measure the model's performance on class 2 test samples containing $z_u$. As illustrated in Fig. 6a, we observe that $S_{\text{sens}}$ for class 3 becomes negative and further decreases, along with degradation in $MP_{2,3}$ as we increase the proportion of $(x_u, y_{L_{\text{fixed}}})$ samples during training. The reason for $S_{\text{sens}}$ for class 3 being negative is that $S_{\text{sens}}$ is calculated with regard to class 3 where $z_u$ was originally inserted in. However, since we have changed the class label to 1 (i.e., $y_{L_{\text{fixed}}}$), $z_u$ starts correlating with class 1, and subsequently $S_{\text{sens}}$ for class 1 increases. Given this shift in the association of $z_u$ towards the fixed noisy label (class 1), we observe **suppression of Feature Memorization**. This phenomenon is supported by the fact that even for very low proportions of $(x_u, y_{L_{\text{fixed}}})$ samples (0.05%-0.36%), the $S_{\text{sens}}$ for class 1 and $MP_{2,1}$ are significantly high, as the model straightaway associates $z_u$ with label 1 as seen in the GradCam plots in Fig. 6c. This associative learning of $z_u$ with the fixed noisy label is exacerbated even further as we increase the proportions of $(x_u, y_{L_{\text{fixed}}})$ samples, which is also evident from the substantial discrepancy in the train-test accuracy presented in Fig. 6b, where the test accuracy for class 2 samples containing $z_u$, drops to nearly 0%, with the proportion of the $(x_u, y_{L_{\text{fixed}}})$ train samples reaching 10%.

These findings clearly verify that **Label Memorization is the root cause of suppressing Feature Memorization and causing Feature Learning**. On top of Fig. 6, results for Medical Abstracts & RoBERTa ($z_u$ = "Riverdale") in Fig. 8 further confirm it, where LM induces FL while suppressing FM, from extremely low proportions of $(x_u, y_{L_{\text{fixed}}})$ and is exacerbated for higher ones. Additional well-aligned results for the remaining datasets UTK-Face, CIFAR10, CIFAR100, NICO++, Emotions

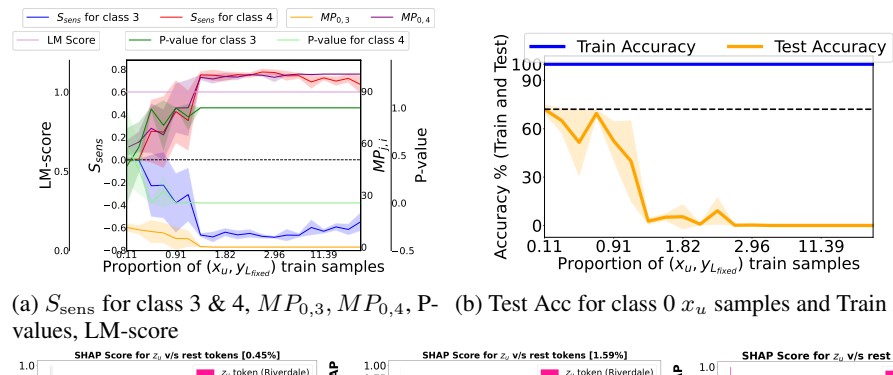

(a) $S_{\text{sens}}$ for class 3 & 4, $MP_{0,3}$, $MP_{0,4}$, P-values, LM-score

(b) Test Acc for class 0 $x_u$ samples and Train Acc

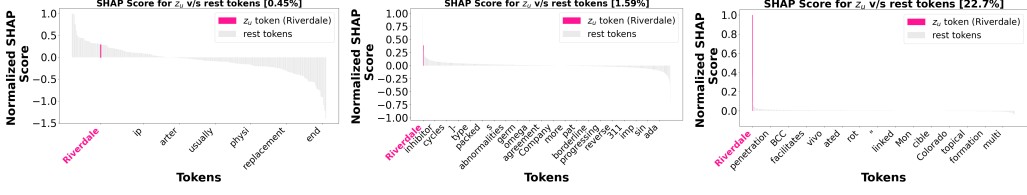

(c) Shapley score results for select $(x_u, y)$ proportions in class 3

Figure 8: **LM causes FL and suppresses FM.** With simultaneous addition of $z_u$ and $y_{L_{\text{fixed}}}$ to the same (image, label) pair, *fixed* LM leads the model to associate $z_u$ with class 4, as evidenced by the increasing $S_{\text{sens}}$ score for class 4. We present vanishing of FM and induced FL even for low proportions of $(x_u, y_{L_{\text{fixed}}})$, supported through high $S_{\text{sens}}$ score, high $MP_{0,4}$, exacerbating train-test accuracy gap, and Shapley score plots where Riverdale gains importance (SHAP score $\gg 0$) even for extremely low proportions. This phenomenon becomes even more intense for higher proportions. (Medical Abstracts, RoBERTa, $z_u$ in class 3 in train, 0 in test & $y_{L_{\text{fixed}}} = 4$)

and model architectures VGG16, ResNet9, GoogleNet, MobileViT, BERT, & RoBERTa, are provided in Appendix C.3.

### 5.2.2 Feature Learning Induced by Label Memorization over Epochs

We understood how Label Memorization causes Feature Learning by making the model associate $z_u$ with $y_{L_{\text{fixed}}}$ in the training phase. Now, we examine the question, **Do the induced Feature Learning and Label Memorization occur in parallel over epochs?** To answer the question, we again look into 3 proportions (1) 0.09%, (2) 0.45%, and (3) 9.09% of $(x_u, y_{L_{\text{fixed}}})$ samples during training in the UTK-Face-ResNet18 setup. We track the progression of the key metrics, $S_{\text{sens}}$, LM-score, p-value, and train-test accuracies over epochs to analyze the relationship between Label Memorization and induced Feature Learning.

### 5.2.3 Observation & Inference

In Fig. 7, we observe that even with an extremely low count of $(x_u, y_{L_{\text{fixed}}})$ samples (0.09%), the model recognizes $z_u$, with LM inducing FL from the early epochs, as evidenced by an increase in the LM-score and a simultaneous increase in $S_{\text{sens}}$ for class 1, which eventually levels off at a very high value. Also, the train-test accuracy gap widens over time, signifying that Feature Learning is caused. This trend becomes even more pronounced as the proportion of $(x_u, y_{L_{\text{fixed}}})$ samples increases further (0.45-9%). We observe a similar progression in the results for additional datasets CIFAR10, CIFAR100, NICO++, & UTK-Face with models ResNet9, GoogleNet MobileViT, & VGG16, as shown in Appendix C.3.

## 6 Conclusion

In this work, we systematically investigate two key questions - (i) Distinguishing Feature Memorization (FM) from Feature Learning (FL) and identifying when they occur, (ii) Investigating whether Label Memorization (LM) influences them. Through extensive experiments across both vision and language tasks, we find that when proportion of samples containing a feature is low, it is memorized, whereas if it is high, it is learned. Moreover, when such features co-occur with noisy labels, LM causes FL and suppresses FM — an effect that starts early in training and plateaus over time. To the best of our knowledge, this is the first work to distinguish FM from FL and reveal their interactions with LM. These findings shed light on the distinct roles of features and labels in deep learning.

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

APPENDIX

# A    TRAINING DETAILS

This section outlines the detailed configurations of the datasets and models used in our experiments, including dataset splits, preprocessing steps, model architectures, and training settings.

## A.1    CIFAR10 AND RESNET9

The **CIFAR10 dataset** (Krizhevsky et al., 2009) is composed of 60,000, 32x32 colored RGB images across 10 different classes (airplane, automobile, bird, cat, deer, dog, frog, horse, ship, truck), with 6,000 images per-class. The dataset is further split into 50,000 training images and 10,000 test images. Before training, we split the dataset into training, validation, and testing set, each comprising 40,000, 10,000, 10,000 images, respectively, equally distributed across all 10 classes.

We employ the **ResNet9** model architecture, a simplified version of the original ResNet (He et al., 2016), with 9 layers in total. It includes 4 convolution blocks, 2 residual blocks, with batch normalization and ReLU activations after each convolution, and followed by max-pooling, flattening, and a classifier layer with 10 output nodes (one for each of the CIFAR10 classes).

## A.2    UTK-FACE AND RESNET18, VGG16

The **UTK-Face dataset** (Zhang et al., 2017) contains over 20,000 face images labeled by age, gender, and ethnicity. For the purpose of our experiments we consider the ethnicity class for the images which has 5 labels in total - White, Asian, Indian, Black, and Others. The original dataset is split into train, validation, and testing sets using a stratified 80:10:10 split. Also, we resize the images to size (224x224x3) for consistency in results.

We use the **ResNet18** (He et al., 2016) & **VGG16** (Simonyan & Zisserman, 2014) architectures, established deep convolutional neural networks designed for efficient feature extraction and classification. The ResNet18 & VGG16 models, are accessed via the torchvision library, and are trained from scratch, with all layers unfrozen to enable fine-tuning. The classifier head is modified to output predictions for the 5 ethnicity classes.

## A.3    CIFAR100 AND GOOGLENET

Similar to CIFAR-10, the **CIFAR-100 dataset** (Krizhevsky et al., 2009) consists of 60,000 32x32 color images, but with 100 different classes, each containing 600 images (500 for training and 100 for testing). We split the dataset into train, validation and test sets, containing 40,000, 10,000 & 10,000 images respectively, equally distributed among all classes.

For the model architecture, we employ the **GoogleNet (Inception v1)** architecture (Szegedy et al., 2015), which utilizes Inception modules to extract multi-scale features. The model architecture is adopted from a publicly available repository [1], while modifying the final classifier layer to output 100 predictions, corresponding to the CIFAR-100 classes.

## A.4    NICO++ AND MOBILEVIT

The **NICO++ dataset** (Zhang et al., 2023) consists of approximately 60,000 images distributed across 60 categories of everyday objects. In this study, we select a subset of 15 object categories, including car, flower, penguin, camel, chair, monitor, truck, wheat, sword, seal, lion, fish, dolphin, lifeboat, and tank. The dataset is partitioned into training (80%), validation (10%), and testing (10%) sets using stratified sampling to ensure balanced representation across all classes. Images are resized to 224x224x3 for consistency with the model input requirements.

We employ the **MobileViT (xxs version)** architecture (Mehta & Rastegari), a lightweight vision transformer designed for efficient processing in resource-constrained environments. The MobileViT

---

[1]https://github.com/weiaicunzai/pytorch-cifar100

model is accessed through the Huggingface timm library[2], and we unfreeze only the last attention block and subsequent layers for fine-tuning. The classifier head is adjusted to output 15 class predictions, corresponding to the selected categories from the NICO++ dataset.

## A.5 MEDICAL ABSTRACTS AND ROBERTA

The **Medical Abstracts dataset**, introduced in Schopf et al. (2022), consists of 14,438 samples with 11,550 in training and 2,888 in testing. We further split the training set using stratified split of 80:20 to get train set and validation set. The abstracts in the dataset consists of 5 (class 0-4) different types of diseases conditions: Neoplasms, Digestive system diseases, Nervous system diseases, Cardiovascular diseases, and General pathological conditions.

We utilize the 12-layered **RoBERTa** model (Liu et al., 2019). We use the Sequence Classification version of the same, available in the Huggingface transformer library[3], with a 5-node output layer for the Emotions dataset.

## A.6 EMOTIONS AND BERT, ROBERTA

The **Emotions dataset**, proposed in Saravia et al. (2018), consists of 16,000 train, 2,000 validation and 2,000 test samples, each sample belonging to one of the 6 classes (class 0-5): sadness, joy, love, anger, fear and surprise. In this paper, we consider only a subset of the train set with 572 samples of each class for faster training.

We utilize the 12-layered **BERT** model (Devlin et al., 2019), a transformer-based language model pre-trained on a large English corpus. We adopt the Sequence Classification version of the same, accessed via the Huggingface Transformers library[4] library. For our experiments, we fine-tune only the last 2 transformer layers and the classification head to adapt to the 6 emotion classes in the Emotions dataset. The classifier head is modified to produce 6 output logits, each corresponding to one of the emotion categories: sadness, joy, love, anger, fear, and surprise. Similarly, we also utilize the 12-layered **RoBERTa** model (Liu et al., 2019), Sequence Classification version, available on Huggingface.

## A.7 TRAINING PARAMETERS

For all the datasets, we consider almost similar training settings. In vision tasks, prior to training, we standardize all images by transforming them to have zero mean and unit variance. This normalization step is essential to ensure stable convergence during training. For the language tasks, we set the max sequence length during tokenization as 512. Furthermore, for both the tasks, we do not use any data augmentations to train our models.

All the models are trained using Stochastic Gradient Descent (SGD) optimizer with a learning rate of 0.1, momentum of 0.9, weight decay of 1e-4, and we also employ a Cosine Annealing learning rate scheduler. For the loss function, we utilize Cross-Entropy Loss, a widely used criterion for multi-class classification tasks. The models are trained for different numbers of epochs depending on the complexity of the dataset and model architecture. Specifically, Emotions with **BERT** and **RoBERTa** is trained for 30 epochs, while Medical Abstracts with RoBERTa is trained for 50 epochs for convergence, **ResNet9**, **GoogleNet**, and **MobileViT** are trained for 50 epochs, and **ResNet18** and **VGG16** are trained for 70 epochs. In all cases, training is continued until convergence, while reaching $\approx$ 100% accuracy on the training set to achieve memorization. For training our models, we utilized Nvidia's A5000ada and A6000 GPUs.

For the per-epochs analysis experiments, we vary the epochs in the following ranges for different datasets:

- **CIFAR-10** : [1, 3, 5, 7, 10, 12, 15, 20, 25, 30, 32, 35, 40, 45, 50]
- **CIFAR-100** : [1, 3, 5, 7, 10, 12, 15, 20, 25, 30, 32, 35, 40, 45, 50]

---

[2]https://huggingface.co/timm
[3]https://huggingface.co/roberta-base
[4]https://huggingface.co/bert-base-uncased

- **UTK-Face** : [1, 5, 10, 15, 20, 30, 32, 25, 40, 45, 50, 55, 60, 65, 70]
- **NICO++** : [1, 3, 5, 7, 10, 12, 15, 20, 25, 30, 32, 35, 40, 45, 50]

## B    FEATURE $z_u$ IN TRAINING

### B.1    TYPES OF FEATURE $z_u$ CONSIDERED

As stated previously, there are no readily available datasets in prior literature that can vary the proportion of a feature in the training dataset. Hence, in this study, we consider 6 different types of features (3 for vision and 3 for language) by introducing them into different datasets. We then track them to distinguish between Feature Memorization and Feature Learning, and their relationship with Label Memorization.

In the vision tasks corresponding to the UTK-Face, CIFAR-10, CIFAR-100, NICO++ datasets, we consider $z_u$ as (i) personal ID ("ID: 452817096712"); (ii) person name ("JOHN DOE"); (iii) English letter ("A"). On the other hand, for the language modality corresponding to the Medical Abstracts and Emotions datasets, we consider $z_u$ as (i) place name ("Riverdale"); (ii-iii) simple words such as "egg" and "ship".

### B.2    ABSOLUTE COUNT CONSISTENCY

In our experiments, we aim to maintain consistency of the count of $z_u$ across all datasets. The absolute count of $z_u$ refers to the number of train samples in which $z_u$ was introduced. This ensures that the experiments are consistent and comparable across datasets. The list of absolute count of $z_u$ that we consider in our experiments for CIFAR10, CIFAR100, UTK-Face is as follows - [0, 1, 2, 4, 6, 8, 10, 12, 14, 16, 18, 20, 25, 30, 35, 40, 60, 100, 200, 300, 400], and for NICO++, we additionally add 500 and 600 to the list. For the Emotions dataset, we follow [0, 1, 2, 4, 6, 8, 10, 12, 14, 16, 20, 40, 45, 50, 60, 65, 70, 80, 85 100, 150, 200, 300, 400, 500] with BERT, and [0, 1, 2, 4, 6, 8, 10, 12, 14, 16, 20, 22, 25, 26, 30, 35, 40, 60, 65, 80, 100, 150, 200, 300, 400, 500] with RoBERTa. Lastly, for the Medical Abstracts dataset we follow - [0, 1, 2, 4, 6, 8, 10, 12, 14, 16, 18, 20, 22, 26, 30, 40, 50, 100, 200, 300, 400], with RoBERTa as the model.

Since datasets vary in the **total number of samples** per class, the same absolute count of $z_u$ naturally results in different ratios across datasets. For instance:

- **CIFAR-10 (Class 0):** With 4000 samples in Class 0, inserting 20 $z_u$ samples results in a 0.5% ratio ($\frac{20}{4000} \times 100$).
- **CIFAR-100 (Class 2):** With 400 samples in Class 2, inserting 20 $z_u$ samples results in a 5% ratio ($\frac{20}{400} \times 100$).
- **UTK-Face (Class 3):** With 2918 samples in Class 3, inserting the same 20 $z_u$ leads to a 0.68% ratio ($\frac{20}{2918} \times 100$).
- **Emotions (Class 4):** With 572 samples in Class 4, inserting $z_u$ in 20 leads to 3.49% ratio.

This approach allows us to avoid biasing the results by adjusting $z_u$ arbitrarily for each dataset. Hence, we systematically scale the relative ratio based on dataset-specific class sizes while keeping the absolute count constant. This ensures **fair representation** of $z_u$ across datasets.

### B.3    GENERALITY OF OBSERVATIONS

Although the ratios differ while keeping the absolute count of $z_u$ the same, the underlying Feature Memorization and Feature Learning phenomena remain consistent across the datasets.

- **At lower concentrations of** $z_u$, Feature Memorization (FM) is observed. Here, the model memorizes the feature without generalizing its utility to the test set.
- **At higher concentrations of** $z_u$, Feature Learning (FL) emerges. The model begins to utilize the learned feature for generalization across test set samples.
- The interplays of FM and FL with Label Memorization (LM) exhibit similar trends across datasets.

For example:

- **CIFAR-10 (Class 0):** Even at a $0.1\%$ ratio (absolute count of $z_u = 4$ in Class 0), FM is observed, while FL becomes dominant at higher ratios like $5\%$ (absolute count of $z_u = 200$ in Class 0).
- **CIFAR-100 (Class 2):** Similar observations are made, where FM is observed for the absolute count of $z_u$ as 4 ($1\%$ ratio) and FL when $z_u$ count is 200 ($50\%$ ratio) in training Class 2.
- **UTK-Face (Class 3):** The same trends are observed for the same absolute counts of $z_u$, with FM seen for $0.18\%$ (absolute count of 4) and FL for $9.1\%$ (absolute count of 200).

These trends suggest that the transition from FM to FL is consistent across the absolute count of $z_u$ in the training class $C_i$, not on the specific size or characteristics of the dataset. Therefore, the observed phenomena are **generalizable and not tied to any particular dataset configuration**.

# C ADDITIONAL EXPERIMENTS & RESULTS

This section presents the technical details of the experimental setups and its corresponding results for additional datasets, previously outlined in Sec 4 and 5, aimed at: (1) establishing a clear distinction between Feature Memorization (FM) and Feature Learning (FL), and (2) analyzing the interaction between Feature Memorization (FM), Feature Learning (FL), and Label Memorization (LM) during the course of training. Additional results regarding the distinction between FM and FL are provided in Sec. C.1, and the results corresponding to influence of LM on FM and FL are present in Sec C.2 & Sec C.3.

## C.1 FM-FL SETUP

In this section, we present the additional results for FM-FL setup for remaining datasets-models configurations, UTK-Face, CIFAR10, CIFAR100, NICO++, Emotions, and models, ResNet18, ResNet9, GoogleNet, MobileViT, BERT, RoBERTa, for varying proportions of $x_u$ in the training set. Furthermore we also provide the plots for how FM and FL varies over epochs for these additional datasets. These results further supports our distinctive definitions of FM and FL, where, if the proportion of samples containing a feature is *low*, the model *memorizes* it, while if the proportion is *high*, the model actually *learns* it.

The results corresponding to UTK-Face, CIFAR10, CIFAR100 NICO++, and Emotions, are presented in Figures 9, 10, 11, 12, 13, 14, 15, 16, 17, & 18.

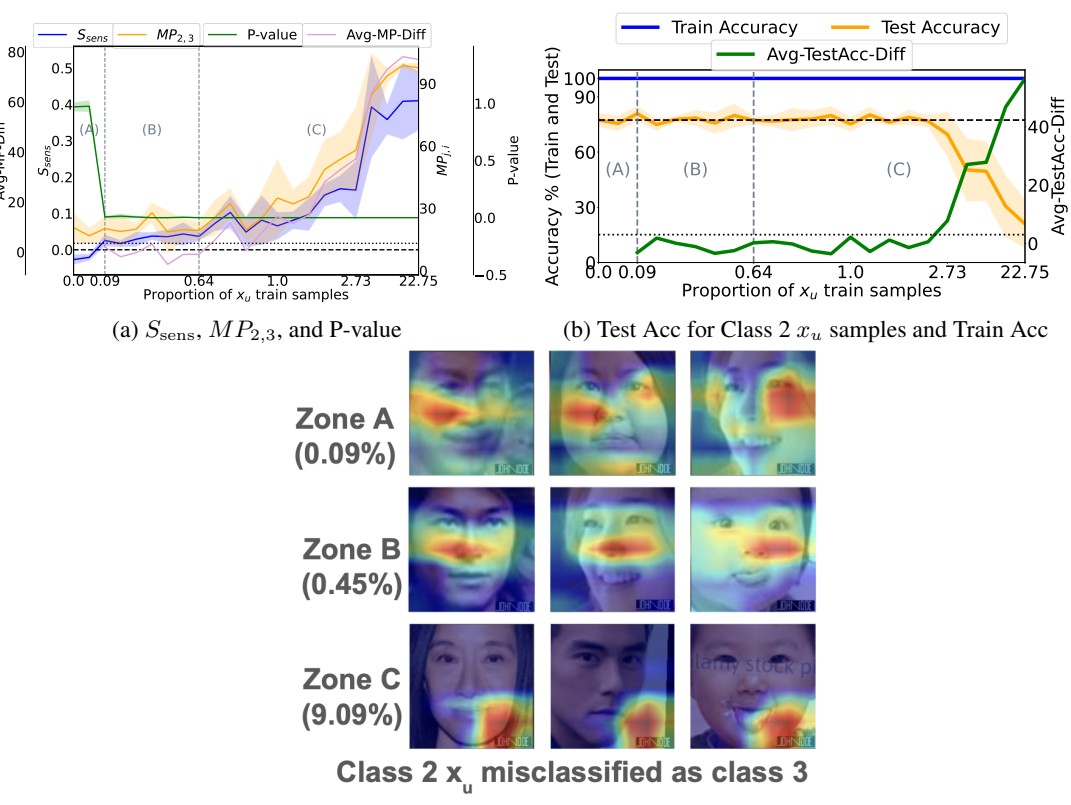

(a) $S_{\text{sens}}$, $MP_{2,3}$, and P-value

(b) Test Acc for Class 2 $x_u$ samples and Train Acc

(c) GradCam results for $(x_u, y)$ ratios in class 3

Figure 9: **FM and shift to FL along with increasing $x_u$ train samples**. We present it using $S_{\text{sens}}$, $MP_{2,3}$, p-value metrics, train-test accuracy gap for $x_u$ train-test samples and the corresponding GradCam results. (UTK-Face, ResNet18 $z_u$ in class 3 in train, 2 in test, Avg-MP-Diff threshold line is at 3.5% (for $\sigma_1$) and Avg-TestAcc-Diff line is at 3% (for $\sigma_2$.)) FM starts at 0.09% (zone (B)) and shifts to FL after 0.64% (zone (C)) of $x_u$ proportion during training, and neither happening for extremely low proportions in zone (A). We do not plot Avg-MP-Diff and Avg-TestAcc-Diff in Zone (A) because the model has not even recognized $z_u$, hence it cannot memorize/learn it.

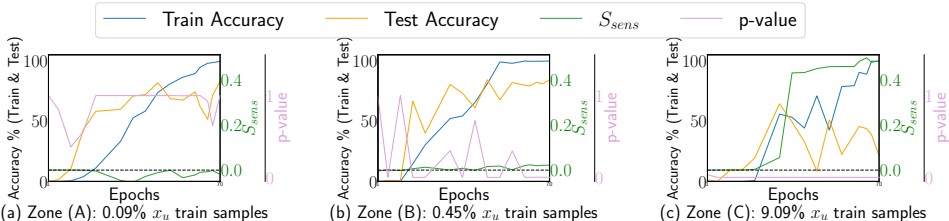

(a) Zone (A): 0.09% $x_u$ train samples    (b) Zone (B): 0.45% $x_u$ train samples    (c) Zone (C): 9.09% $x_u$ train samples

Figure 10: **Analysis of FM & FL over epochs.** We explore how FM & FL occurs over the course of training by measuring $S_{\text{sens}}$, p-value, and train-test accuracy gap of $x_u$ train and test samples over epochs. (UTK-Face, ResNet18, $z_u$ in class 3 in train, 2 in test)

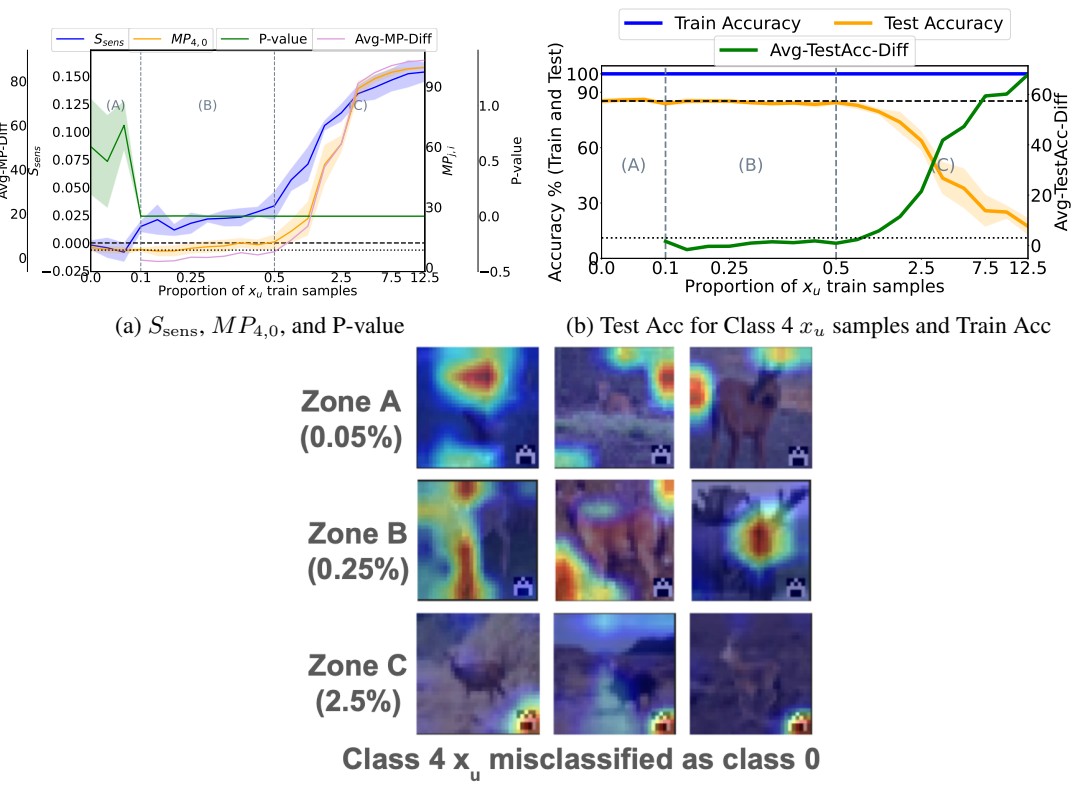

(a) $S_{\text{sens}}$, $MP_{4,0}$, and P-value

(b) Test Acc for Class 4 $x_u$ samples and Train Acc

(c) GradCam results for $(x_u, y)$ ratios in class 0

Figure 11: **FM and shift to FL along with increasing $x_u$ train samples**. We present it using $S_{\text{sens}}$, $MP_{4,0}$, p-value metrics, train-test accuracy gap for $x_u$ train-test samples and the corresponding GradCam results. (CIFAR10, ResNet9, $z_u$ in class 0 in train, 4 in test, Avg-MP-Diff threshold line is at 3.5% (for $\sigma_1$) and Avg-TestAcc-Diff line is at 3% (for $\sigma_2$.)) FM starts at 0.1% (zone (B)) and shifts to FL after 0.5% (zone (C)) of $x_u$ proportion during training, and neither happening for extremely low proportions in zone (A). We do not plot Avg-MP-Diff and Avg-TestAcc-Diff in Zone (A) because the model has not even recognized $z_u$, hence it cannot memorize/learn it.

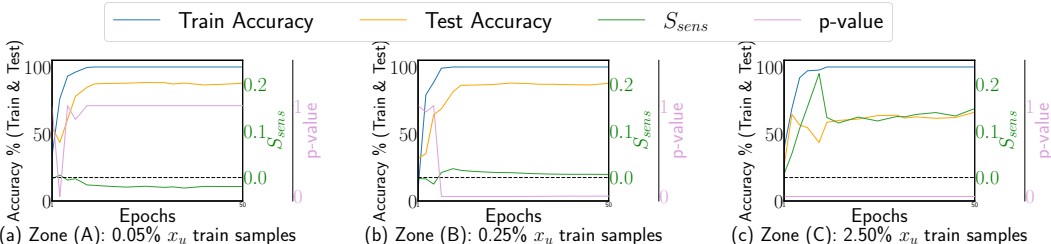

(a) Zone (A): 0.05% $x_u$ train samples

(b) Zone (B): 0.25% $x_u$ train samples

(c) Zone (C): 2.50% $x_u$ train samples

Figure 12: **Analysis of FM & FL over epochs.** We explore how FM & FL occurs over the course of training by measuring $S_{\text{sens}}$, p-value, and train-test accuracy gap of $x_u$ train and test samples over epochs. (CIFAR10, ResNet9, $z_u$ in class 0 in train, 4 in test)

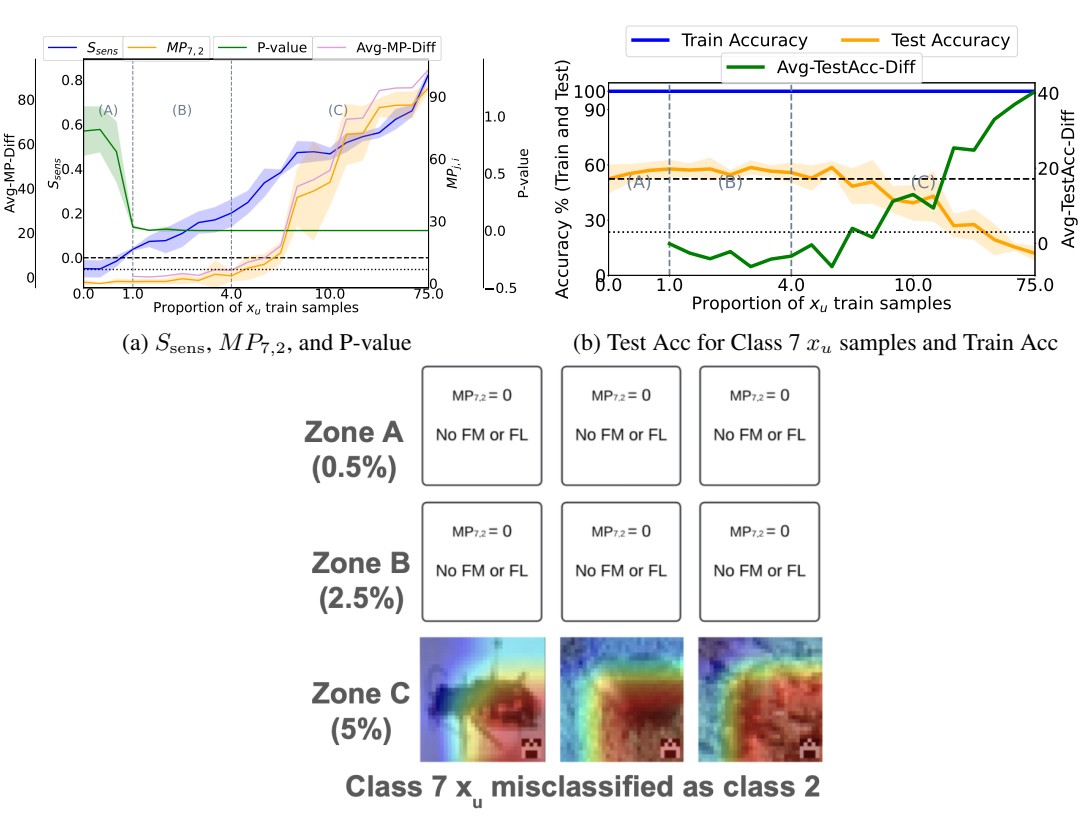

(a) $S_{\text{sens}}$, $MP_{7,2}$, and P-value

(b) Test Acc for Class 7 $x_u$ samples and Train Acc

(c) GradCam results for $(x_u, y)$ ratios in class 2

Figure 13: **FM and shift to FL along with increasing $x_u$ train samples**. We present it using $S_{\text{sens}}$, $MP_{7,2}$, p-value metrics, train-test accuracy gap for $x_u$ train-test samples and the corresponding GradCam results. (CIFAR100, GoogleNet, $z_u$ in class 2 in train, 7 in test, Avg-MP-Diff threshold line is at 3.5% (for $\sigma_1$) and Avg-TestAcc-Diff line is at 3% (for $\sigma_2$.)) FM starts at 1% (zone (B)) and shifts to FL after 4% (zone (C)) of $x_u$ proportion during training, with neither happening for extremely low proportions in zone (A). We do not provide GradCam plots for test images for 0.5% and 2.5%, because we don't observe any $MP_{7,2}$ for both the cases - which aligns with the definition of FM (2.5% proportion). We do not plot Avg-MP-Diff and Avg-TestAcc-Diff in Zone (A) because the model has not even recognized $z_u$, hence it cannot memorize/learn it.

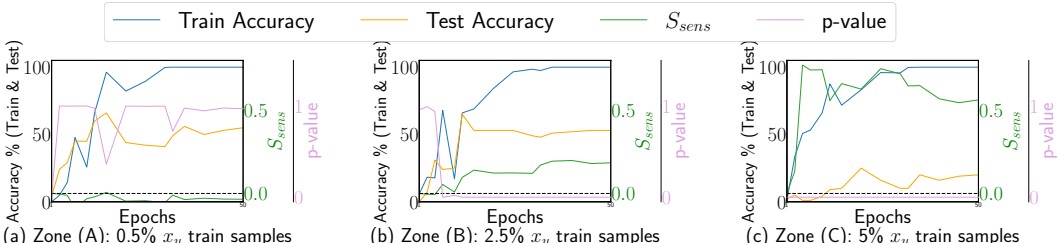

(a) Zone (A): 0.5% $x_u$ train samples  (b) Zone (B): 2.5% $x_u$ train samples  (c) Zone (C): 5% $x_u$ train samples

Figure 14: **Analysis of FM & FL over epochs.** We explore how FM & FL occurs over the course of training by measuring $S_{\text{sens}}$, p-value, and train-test accuracy gap of $x_u$ train and test samples over epochs. (CIFAR100, GoogleNet, $z_u$ in class 2 in train, 7 in test)

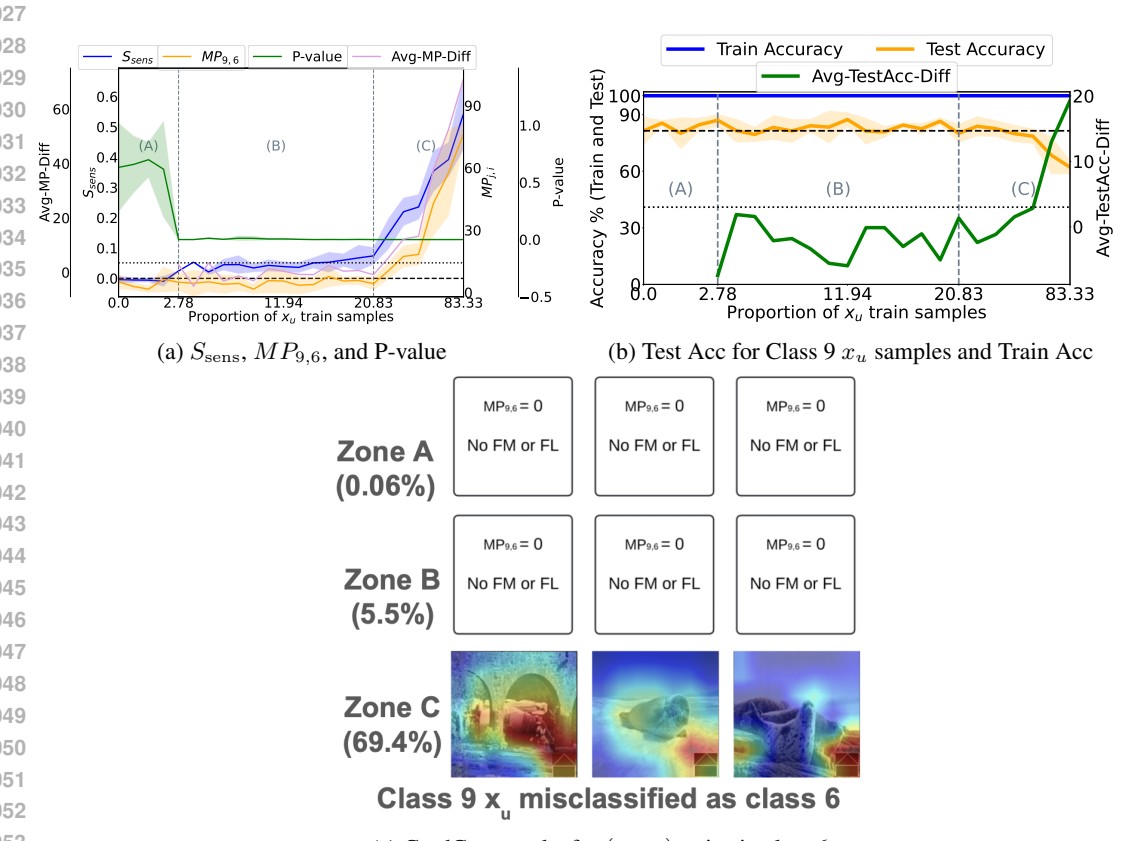

(a) $S_{\text{sens}}$, $MP_{9,6}$, and P-value

(b) Test Acc for Class 9 $x_u$ samples and Train Acc

(c) GradCam results for $(x_u, y)$ ratios in class 6

Figure 15: **FM and shift to FL along with increasing $x_u$ train samples**. We present it using $S_{\text{sens}}$, $MP_{9,6}$, p-value metrics, train-test accuracy gap for $x_u$ train-test samples and the corresponding GradCam results. (NICO++, MobileViT, $z_u$ in class 6 in train, 9 in test, Avg-MP-Diff threshold line is at 3.5% (for $\sigma_1$) and Avg-TestAcc-Diff line is at 3% (for $\sigma_2$.)) FM starts at 2.78% (zone (B)) and shifts to FL after 20.83% (zone (C)) of $x_u$ proportion during training, with neither happening for extremely low proportions in zone (A). We do not provide GradCam plots for test images for 0.06% and 5.5%, because we do not observe any $MP_{9,6}$ for both the cases - which aligns with the definition of FM (5.5% proportion). We do not plot Avg-MP-Diff and Avg-TestAcc-Diff in Zone (A) because the model has not even recognized $z_u$, hence it cannot memorize/learn it.

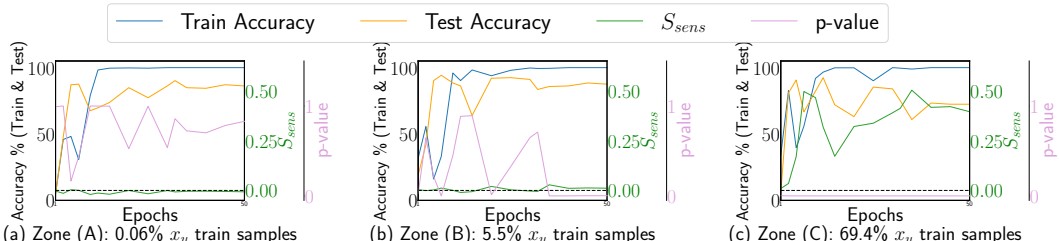

(a) Zone (A): 0.06% $x_u$ train samples
(b) Zone (B): 5.5% $x_u$ train samples
(c) Zone (C): 69.4% $x_u$ train samples

Figure 16: **Analysis of FM & FL over epochs.** We explore how FM & FL occurs over the course of training by measuring $S_{\text{sens}}$, p-value, and train-test accuracy gap of $x_u$ train and test samples over epochs. (NICO++, MobileViT, $z_u$ in class 6 in train, 9 in test)

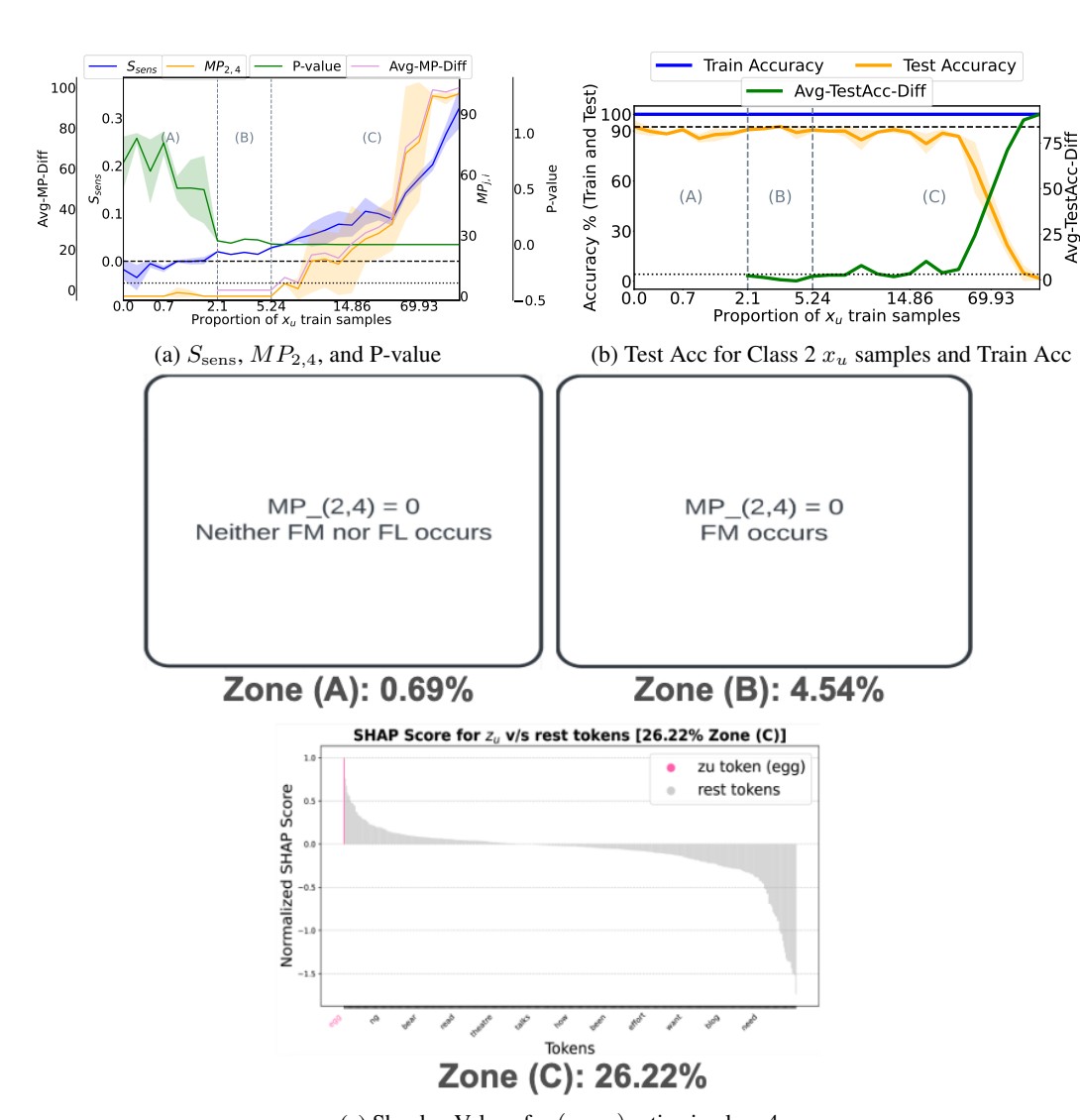

(a) $S_{\text{sens}}$, $MP_{2,4}$, and P-value

(b) Test Acc for Class 2 $x_u$ samples and Train Acc

(c) Shapley Values for $(x_u, y)$ ratios in class 4

Figure 17: **FM and shift to FL with increasing $x_u$ train samples**. We present it using $S_{\text{sens}}$, $MP_{2,4}$, p-value, train-test accuracy gap for $x_u$ train-test samples, and Shapley-Value results, with Avg-MP-Diff threshold: 3.5% ($\sigma_1$), Avg-TestAcc-Diff: 3% ($\sigma_2$). FM starts at 2.1% (zone B) and shifts to FL after 5.24% (zone C), with neither occurring at very low proportions (zone A). Shapley values confirm these transitions, where for Zone (B) of FM, "egg"-bar plot has negligible importance (SHAP score $\approx 0$) in testing even though it has been recognized, while for Zone (C) of FL, "egg" gains importance (SHAP score $\gg 0$), showing generalization of "egg" to the test set. (Emotions, BERT, $z_u$ in class 4 in train, 2 in test). We do not plot Avg-MP-Diff and Avg-TestAcc-Diff in Zone (A) because the model has not even recognized $z_u$, hence it cannot memorize/learn it.

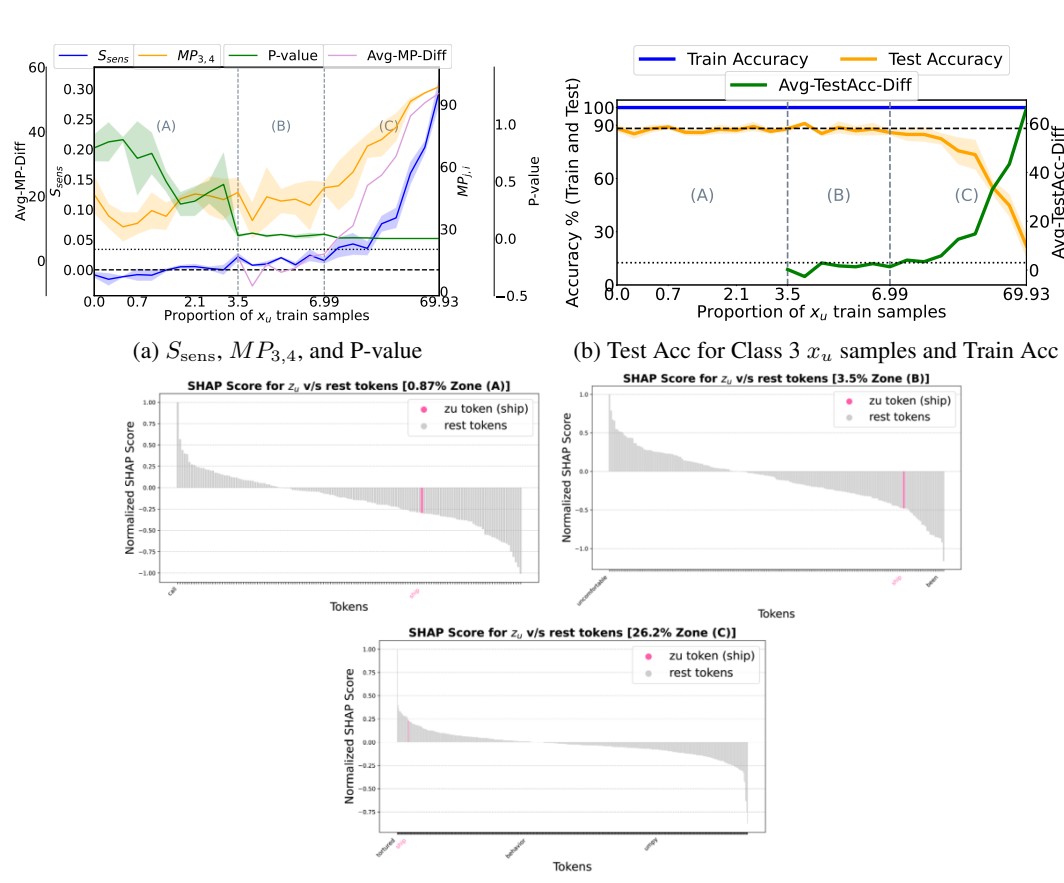

(a) $S_{\text{sens}}$, $MP_{3,4}$, and P-value

(b) Test Acc for Class 3 $x_u$ samples and Train Acc

(c) Shapley Values for $(x_u, y)$ ratios in class 4

Figure 18: **FM and shift to FL with increasing $x_u$ train samples**. We present it using $S_{\text{sens}}$, $MP_{3,4}$, p-value, train-test accuracy gap for $x_u$ train-test samples, and Shapley-Value results, with Avg-MP-Diff threshold: 3.5% ($\sigma_1$), Avg-TestAcc-Diff: 3% ($\sigma_2$). FM starts at 3.5% (zone B) and shifts to FL after 6.99% (zone C), with neither occurring at very low proportions (zone A). Shapley values confirm these transitions, where for Zone (B) of FM, "ship" bar plot has negligible importance (SHAP score $< 0$) in testing even though it has been recognized, while for Zone (C) of FL, "ship" gains importance (SHAP score $\gg 0$), showing generalization of "ship" to the test set. (Emotions, RoBERTa, $z_u$ in class 4 in train, 3 in test). We do not plot Avg-MP-Diff and Avg-TestAcc-Diff in Zone (A) because the model has not even recognized $z_u$, hence it cannot memorize/learn it.

## C.2 JOINT FEATURE AND *Random* NOISY LABEL SETUP

This section covers the results for the remaining datasets, CIFAR10, CIFAR100, UTK-Face, Medical Abstracts, Emotions, and models, ResNet9, GoogleNet, VGG16, ResNet18, BERT, RoBERTa, for increasing proportions of $(x_u, y_L)$ in the training set. The results for all these datasets, models are presented in Figures 19, 20, 21, 22, 23, 24, and 25.

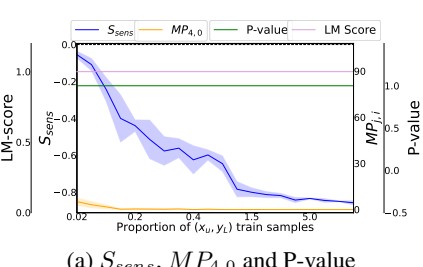
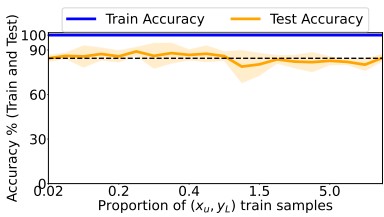

(a) $S_{sens}$, $MP_{4,0}$ and P-value

(b) Test Acc for class 4 $x_u$ samples and Train Acc

Figure 19: **Disappearance of FM and FL.** As $(x_u, y_L)$ training samples increases, $S_{\text{sens}}$ becomes negative and further decreases, while $MP_{4,0}$ stays minimal. The train-test accuracy gap for $x_u$ samples remains stable, indicating the disappearance of FM and FL, while *random* LM is still present.(CIFAR10, ResNet9, $z_u$ in class 0 in train, 4 in test)

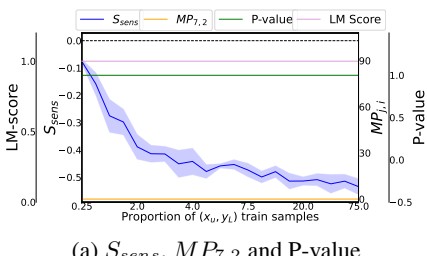
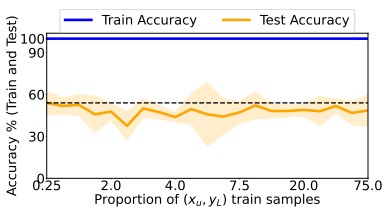

(a) $S_{sens}$, $MP_{7,2}$ and P-value

(b) Test Acc for class 7 $x_u$ samples and Train Acc

Figure 20: **Disappearance of FM and FL.** As $(x_u, y_L)$ training samples increases, $S_{\text{sens}}$ becomes negative and further decreases, while $MP_{7,2}$ stays minimal. The train-test accuracy gap for $x_u$ samples remains stable, indicating the disappearance of FM and FL, while *random* LM is still present. (CIFAR100, GoogleNet, $z_u$ in class 2 in train, 7 in test)

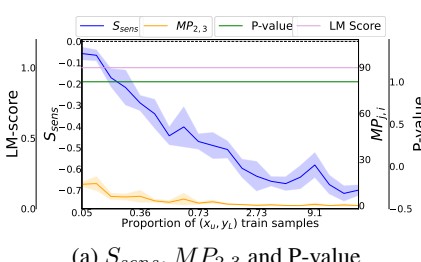
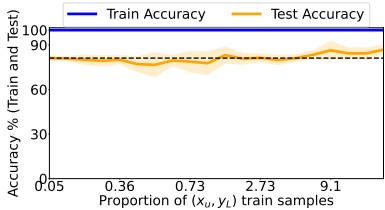

(a) $S_{sens}$, $MP_{2,3}$ and P-value

(b) Test Acc for class 2 $x_u$ samples and Train Acc

Figure 21: **Disappearance of FM and FL.** As $(x_u, y_L)$ training samples increases, $S_{\text{sens}}$ becomes negative and further decreases, while $MP_{2,3}$ stays minimal. The train-test accuracy gap for $x_u$ samples remains stable, indicating the disappearance of FM and FL, while *random* LM is still present.(UTK-Face, ResNet18, $z_u$ in class 3 in train, 2 in test)

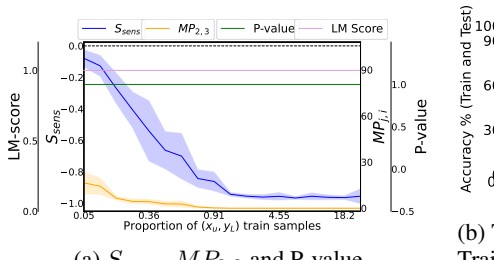
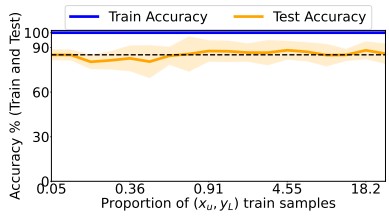

(a) $S_{sens}$, $MP_{2,3}$ and P-value

(b) Test Acc for class 2 $x_u$ samples and Train Acc

Figure 22: **Disappearance of FM and FL.** As $(x_u, y_L)$ training samples increases, $S_{\text{sens}}$ becomes negative and further decreases, while $MP_{2,3}$ stays minimal. The train-test accuracy gap for $x_u$ samples remains stable, indicating the disappearance of FM and FL, while *random* LM is still present.(UTK-Face, VGG16, $z_u$ in class 3 in train, 2 in test)

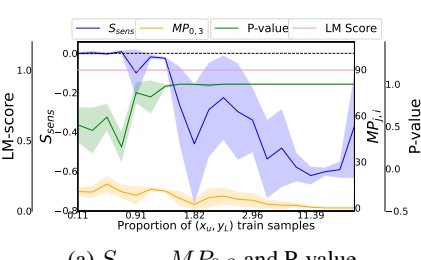
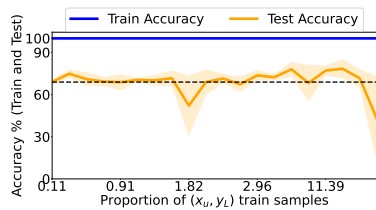

(a) $S_{sens}$, $MP_{0,3}$ and P-value

(b) Test Acc for class 0 $x_u$ samples and Train Acc

Figure 23: **Disappearance of FM and FL.** As $(x_u, y_L)$ training samples increases, $S_{\text{sens}}$ becomes negative and further decreases, while $MP_{0,3}$ stays minimal. The train-test accuracy gap for $x_u$ samples remains stable, indicating the disappearance of FM and FL, while *random* LM is still present. (Medical Abstracts, RoBERTa, $z_u$ in class 3 in train, 0 in test).

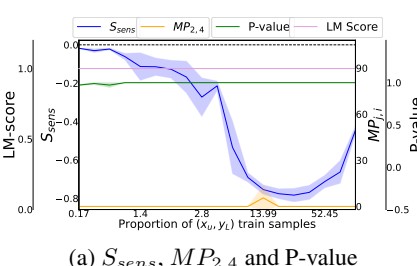
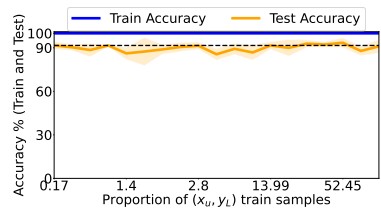

(a) $S_{sens}$, $MP_{2,4}$ and P-value

(b) Test Acc for class 2 $x_u$ samples and Train Acc

Figure 24: **Disappearance of FM and FL.** As $(x_u, y_L)$ training samples increases, $S_{\text{sens}}$ becomes negative and further decreases, while $MP_{2,4}$ stays minimal. The train-test accuracy gap for $x_u$ samples remains stable, indicating the disappearance of FM and FL, while *random* LM is still present. (Emotions, BERT, $z_u$ in class 4 in train, 2 in test).

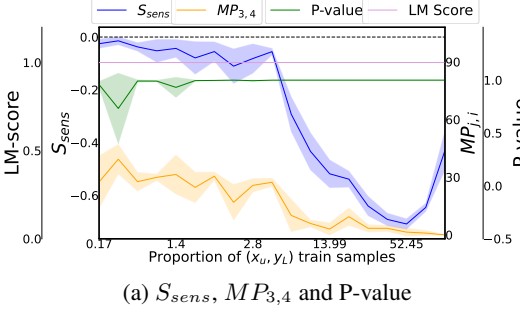
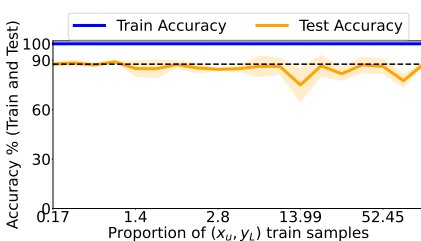

(a) $S_{sens}$, $MP_{3,4}$ and P-value

(b) Test Acc for class 3 $x_u$ samples and Train Acc

Figure 25: **Disappearance of FM and FL.** As $(x_u, y_L)$ training samples increases, $S_{\text{sens}}$ becomes negative and further decreases, while $MP_{3,4}$ stays minimal. The train-test accuracy gap for $x_u$ samples remains stable, indicating the disappearance of FM and FL, while *random* LM is still present. (Emotions, RoBERTa, $z_u$ in class 4 in train, 3 in test).

### C.3 JOINT FEATURE AND *Fixed* NOISY LABEL SETUP

In this section, we provide additional results across the remaining datasets-models configurations, for CIFAR10, CIFAR100, UTK-Face, NICO++, Medical Abstracts, Emotions, corresponding to models, ResNet9, GoogleNet, VGG16, MobileViT, BERT, RoBERTa, while varying the proportions of $(x_u, y_{L_{\text{fixed}}})$ in the training set. Furthermore, we also provide additional plots on how the induced Feature Learning phenomenon goes in parallel with Label Memorization over the course of training. These results on remaining datasets, models further confirms our claim that ***Label Memorization causes Feature Learning*** and ***suppresses Feature Memorization.*** The results corresponding to CIFAR10 & ResNet9, UTK-Face & VGG16, CIFAR100 & GoogleNet, NICO++ & MobileViT, and Emotions & BERT, RoBERTa, are presented in Figs. 26, 27, 28, 29, 30, 31, 32, 33, 34, and 35, respectively.

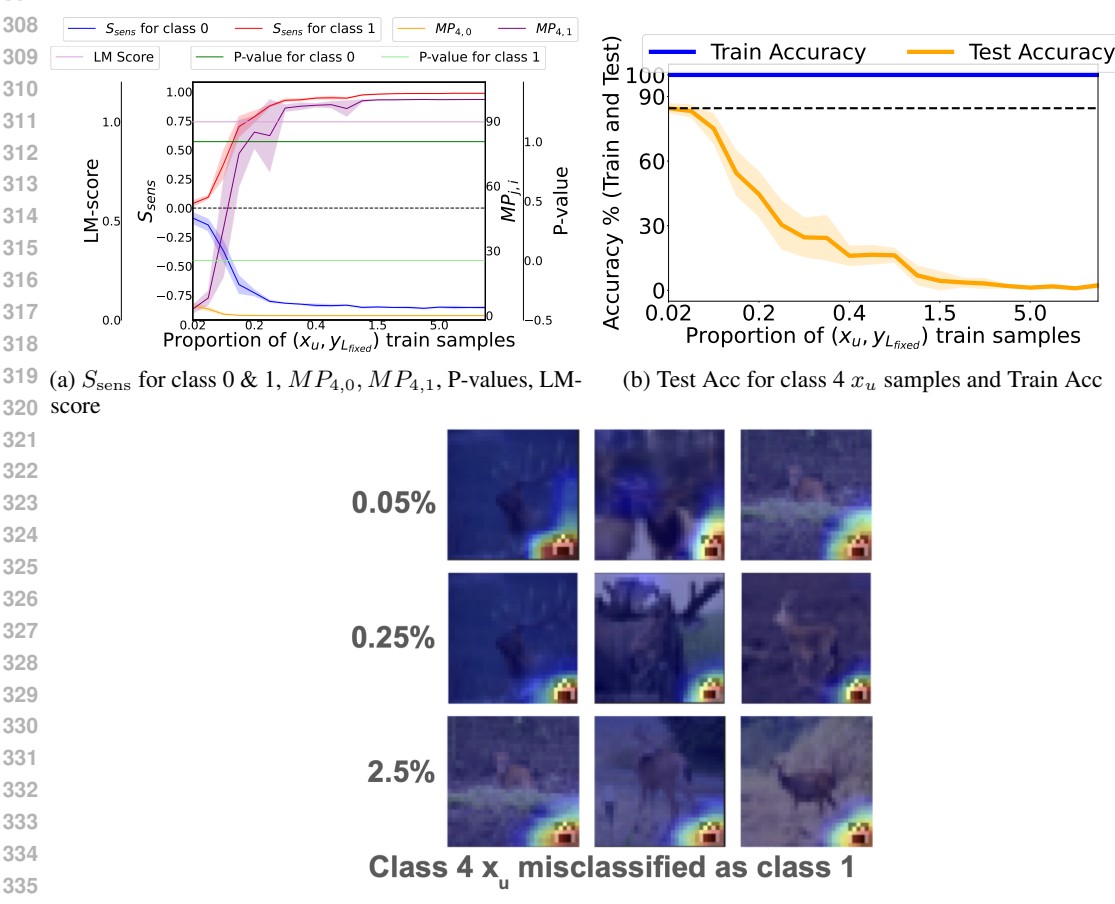

(a) $S_{\text{sens}}$ for class 0 & 1, $MP_{4,0}$, $MP_{4,1}$, P-values, LM-score

(b) Test Acc for class 4 $x_u$ samples and Train Acc

(c) GradCam results for $(x_u, y_{L_{\text{fixed}}})$ ratios in class 0

Figure 26: **LM causes FL and suppresses FM.** With simultaneous addition of $z_u$ and $y_{L_{\text{fixed}}}$ to the same (image, label) pair, LM leads the model to associate $z_u$ with class 1, as evidenced by the increasing $S_{\text{sens}}$ score for class 1. We present vanishing of FM and induced FL even for low proportions of $(x_u, y_{L_{\text{fixed}}})$, supported through high $S_{\text{sens}}$ score, high $MP_{4,1}$, exacerbating train-test accuracy gap, and GradCam plots. This phenomenon becomes even more intense for higher proportions. (CIFAR10, ResNet9, $z_u$ in class 0 in train, 4 in test & $y_{L_{\text{fixed}}} = 1$)

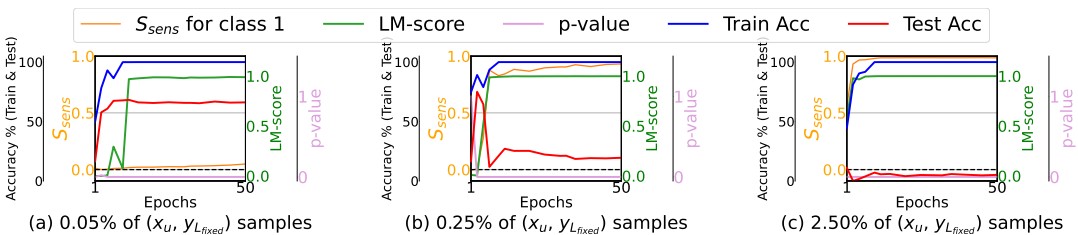

(a) 0.05% of $(x_u, y_{L_{fixed}})$ samples  (b) 0.25% of $(x_u, y_{L_{fixed}})$ samples  (c) 2.50% of $(x_u, y_{L_{fixed}})$ samples

Figure 27: **Induced FL and LM occur in parallel over epochs.** We observe that when $z_u$ and $y_{L_{fixed}}$ are in the same (image, label) pair, then LM causes FL concurrently over epochs even for extremely low concentration of $z_u$ in training. This observation is also supported by increasing LM score, increasing $S_{\text{sens}}$ score and widened train-test accuracy gap as training progresses. (CIFAR10, ResNet9, $z_u$ in class 0 in train, 4 in test & $y_{L_{fixed}} = 1$)

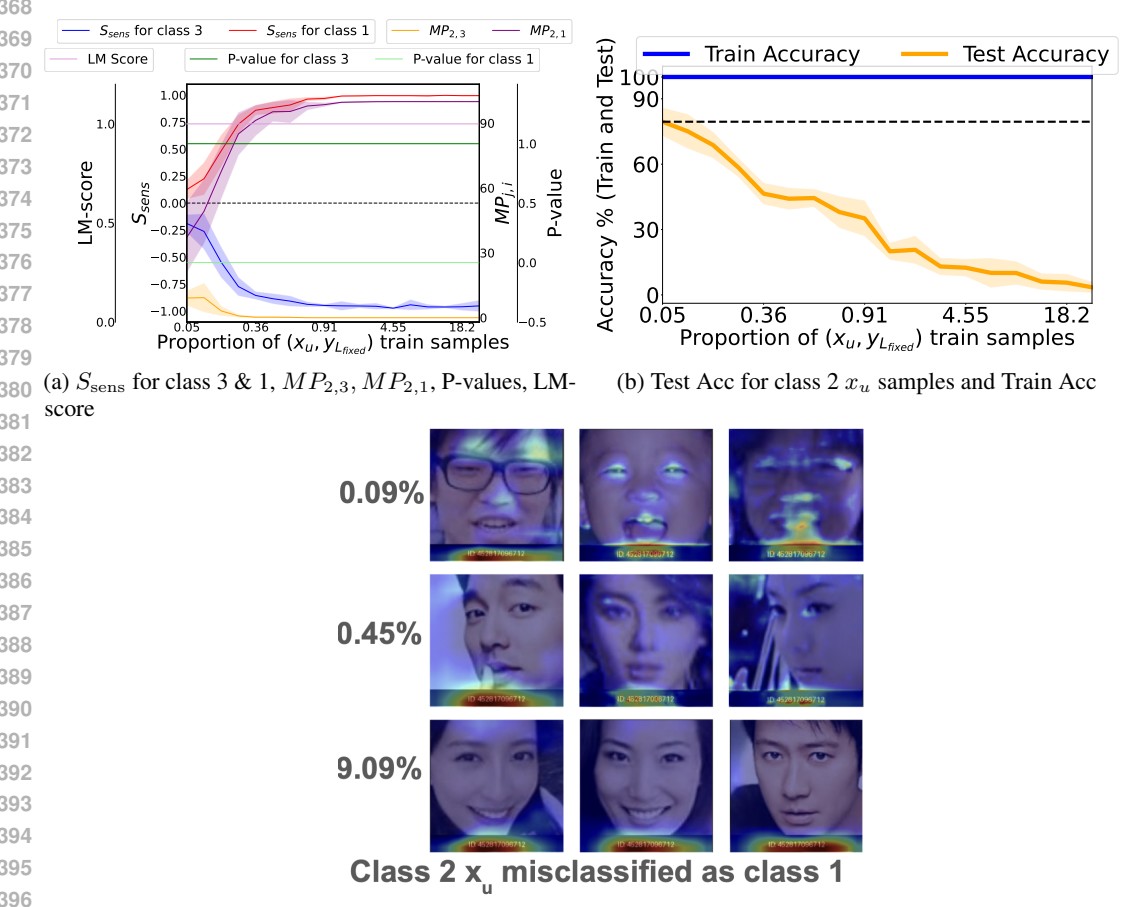

(a) $S_{\text{sens}}$ for class 3 & 1, $MP_{2,3}$, $MP_{2,1}$, P-values, LM-score

(b) Test Acc for class 2 $x_u$ samples and Train Acc

(c) GradCam results for $(x_u, y_{L_{fixed}})$ ratios in class 3

Figure 28: **LM causes FL and suppresses FM.** With concurrent addition of $z_u$ and $y_{L_{fixed}}$, *fixed* LM leads the model to associate $z_u$ with class 1, as evidenced by the increasing $S_{\text{sens}}$ score for class 1. It presents vanishing of FM and induced FL even for low proportions of $(x_u, y_{L_{fixed}})$, supported by high $S_{\text{sens}}$ score, high $MP_{2,1}$, wider train-test accuracy gap, and GradCam.(UTK-Face, VGG16, $z_u$ in class 3 in train, 2 in test & $y_{L_{fixed}} = 1$)

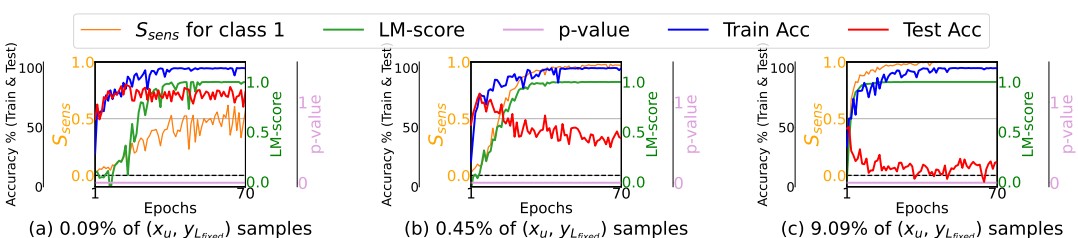

(a) 0.09% of $(x_u, y_{L_{fixed}})$ samples    (b) 0.45% of $(x_u, y_{L_{fixed}})$ samples    (c) 9.09% of $(x_u, y_{L_{fixed}})$ samples

Figure 29: **Induced FL and LM occur in parallel over epochs.** We observe that when $z_u$ and $y_{L_{fixed}}$ are in the same (image, label) pair, LM causes FL concurrently over epochs even for extremely low concentration of $z_u$ in training. This is supported by increasing LM score and $S_{sens}$ score, and widening train-test accuracy gap as training progresses. (UTK-Face, VGG16, $z_u$ in class 3 in train, 2 in test & $y_{L_{fixed}} = 1$)

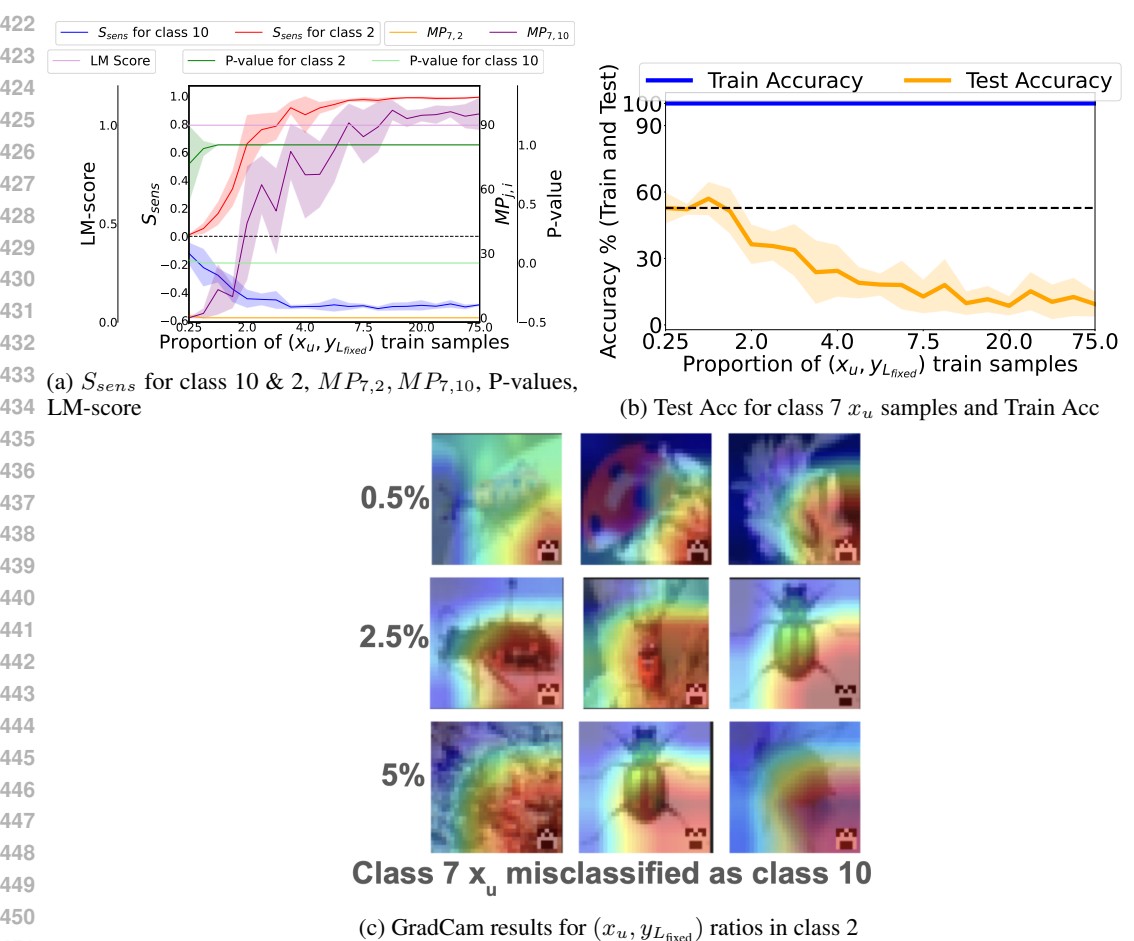

(a) $S_{sens}$ for class 10 & 2, $MP_{7,2}, MP_{7,10}$, P-values, LM-score

(b) Test Acc for class 7 $x_u$ samples and Train Acc

(c) GradCam results for $(x_u, y_{L_{fixed}})$ ratios in class 2

Figure 30: **LM causes FL and suppresses FM.** With concurrent addition of $z_u$ and $y_{L_{fixed}}$, LM leads the model to associate $z_u$ with class 10, as evidenced by the increasing $S_{sens}$ score for class 10. It presents vanishing of FM and induced FL even for low proportions of $(x_u, y_{L_{fixed}})$, supported by high $S_{sens}$ score, high $MP_{7,10}$, wider train-test accuracy gap, and GradCam.(CIFAR100, GoogleNet, $z_u$ in class 2 in train, 7 in test & $y_{L_{fixed}} = 10$)

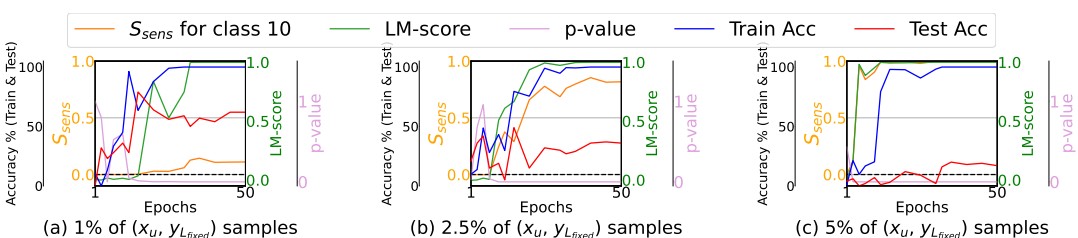

(a) 1% of $(x_u, y_{L_{fixed}})$ samples          (b) 2.5% of $(x_u, y_{L_{fixed}})$ samples          (c) 5% of $(x_u, y_{L_{fixed}})$ samples

Figure 31: **Induced FL and LM occur parallely over epochs.** We observe that when $z_u$ and $y_{L_{fixed}}$ are in the same (image, label) pair, then LM causes FL concurrently over epochs even for extremely low concentration of $z_u$ in training. This observation is also supported by increasing LM score, increasing $S_{sens}$ score, and widening of train-test accuracy gap as training progresses.(CIFAR100, GoogleNet, $z_u$ in class 2 in train, 7 in test & $y_{L_{fixed}}$= 10)

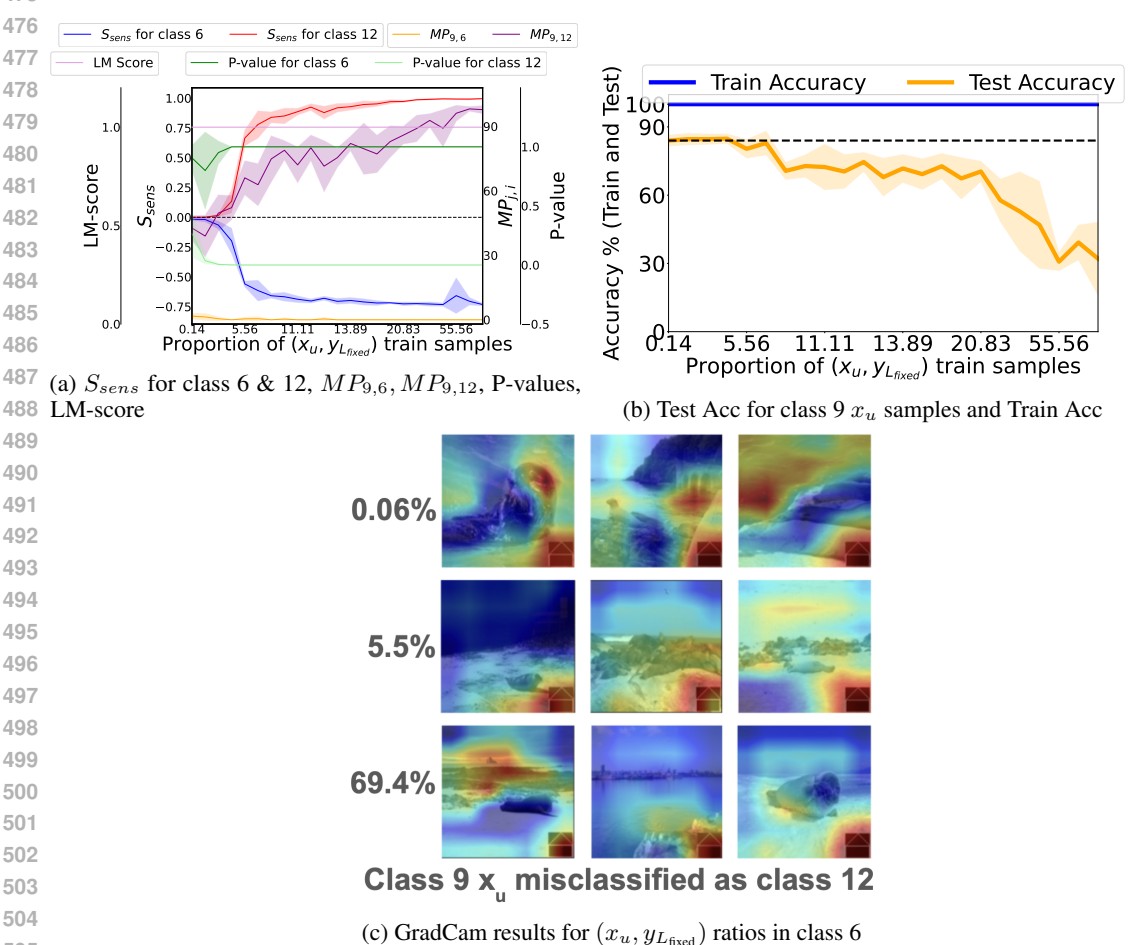

(a) $S_{sens}$ for class 6 & 12, $MP_{9,6}$, $MP_{9,12}$, P-values, LM-score

(b) Test Acc for class 9 $x_u$ samples and Train Acc

(c) GradCam results for $(x_u, y_{L_{fixed}})$ ratios in class 6

Figure 32: **LM causes FL and suppresses FM.** With concurrent addition of $z_u$ and $y_{L_{fixed}}$ , LM leads the model to associate $z_u$ with class 12, as evidenced by the increasing $S_{sens}$ score for class 12. It presents vanishing of FM and induced FL even for low proportions of $(x_u, y_{L_{fixed}})$, supported by high $S_{sens}$ score, high $MP_{9,12}$, wider train-test accuracy gap, and GradCam.(NICO++, MobileViT, $z_u$ in class 6 in train, 9 in test & $y_{L_{fixed}}$=12)

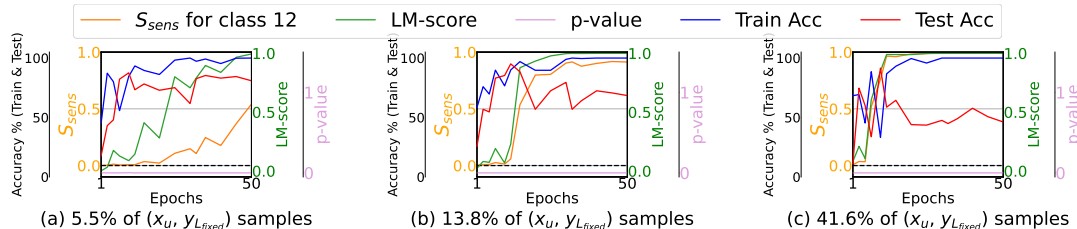

(a) 5.5% of $(x_u, y_{L_{fixed}})$ samples   (b) 13.8% of $(x_u, y_{L_{fixed}})$ samples   (c) 41.6% of $(x_u, y_{L_{fixed}})$ samples

Figure 33: **Induced FL and LM occur parallely over epochs.** We observe that when $z_u$ and $y_{L_{fixed}}$ are in the same (image, label) pair, then LM causes FL concurrently over epochs even for extremely low concentration of $z_u$ in training. This observation is also supported by increasing LM score, increasing $S_{sens}$ score and widening of train-test accuracy gap as training progresses. (NICO++, MobileViT, $z_u$ in class 6 in train, 9 in test & $y_{L_{fixed}}$=12)

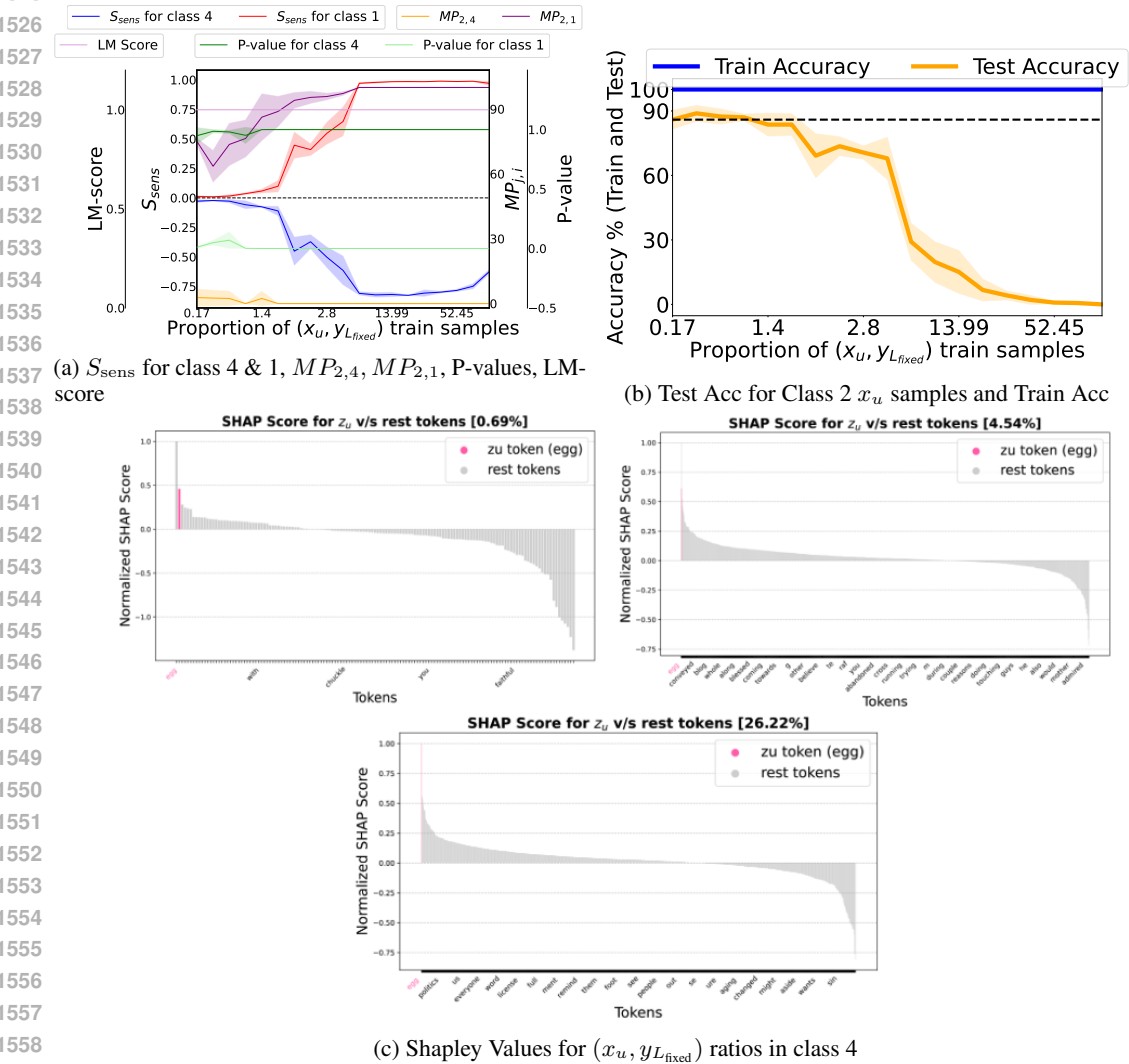

(a) $S_{sens}$ for class 4 & 1, $MP_{2,4}$, $MP_{2,1}$, P-values, LM-score

(b) Test Acc for Class 2 $x_u$ samples and Train Acc

(c) Shapley Values for $(x_u, y_{L_{fixed}})$ ratios in class 4

Figure 34: **LM causes FL and suppresses FM.** With simultaneous addition of $z_u$ and $y_{L_{fixed}}$ to the same (image, label) pair, LM leads the model to associate $z_u$ with class 1, as evidenced by the increasing $S_{sens}$ score for class 1. We present LM causing FL and suppressing FM even for low proportions of $(x_u, y_{L_{fixed}})$, supported through high $S_{sens}$ score, high $MP_{2,1}$, exacerbating train-test accuracy gap. The Shapley plots further confirm this, where even for low proportions of 0.69%, "egg" bar plot gains importance (SHAP score $\gg 0$) in the test-time predictions. This phenomenon becomes even more intense for higher proportions.(Emotions, BERT, $z_u$ in class 4 in train, 2 in test & $y_{L_{fixed}}$=1)

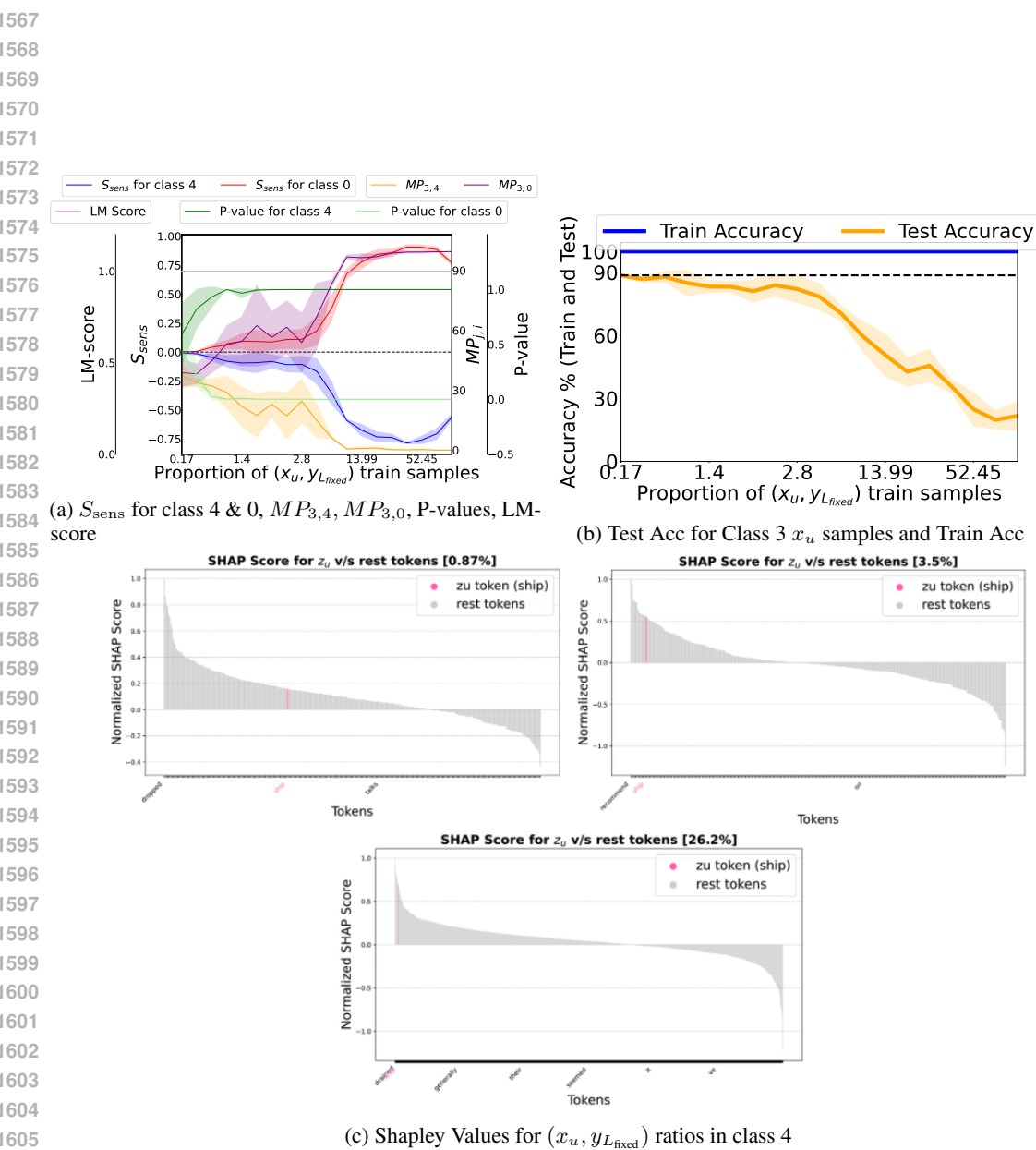

(a) $S_{\text{sens}}$ for class 4 & 0, $MP_{3,4}$, $MP_{3,0}$, P-values, LM-score

(b) Test Acc for Class 3 $x_u$ samples and Train Acc

(c) Shapley Values for $(x_u, y_{L_{\text{fixed}}})$ ratios in class 4

Figure 35: **LM causes FL and suppresses FM.** With simultaneous addition of $z_u$ and $y_{L_{\text{fixed}}}$ to the same (image, label) pair, LM leads the model to associate $z_u$ with class 0, as evidenced by the increasing $S_{\text{sens}}$ score for class 0. We present LM causing FL and suppressing FM even for low proportions of $(x_u, y_{L_{\text{fixed}}})$, supported through high $S_{\text{sens}}$ score, high $MP_{3,0}$, exacerbating train-test accuracy gap. The Shapley plots further confirm this, where even for low proportions of 0.87%, "ship" bar plot gains importance (SHAP score $\gg 0$) in the test-time predictions. This phenomenon becomes even more intense for higher proportions.(Emotions, RoBERTa, $z_u$ in class 4 in train, 3 in test & $y_{L_{\text{fixed}}}$=0)

# D   DEFINING THRESHOLDS ($\sigma_1$ AND $\sigma_2$) FOR DISTINGUISHING FM AND FL

The results presented in our paper reveal that we can robustly compute $\sigma_1$ and $\sigma_2$, which remain consistent across all the datasets, aligning well with our findings. Please note that our experiments are run across 5 seeds. For every dataset, we compute the **Avg $MP_{j,i}$ (%)** and **Avg Test Accuracy of class $C_j$ samples (%)** containing $z_u$. These are calculated across increasing ratios of $x_u$ training samples containing $z_u$.

After computing them, we find the **Avg-MP-Diff (%)** and **Avg-TestAcc-Diff (%)** scores as defined in Sec. 3, for varying ratios (%) of training samples containing $z_u$ for class $C_i$.

We provide tabular results for all 8 datasets-models configurations across vision and language modalities, and analyze them one by one: (Wherever we see N/A in the tables, it means neither FM nor FL occurs, i.e., the model has not recognized the feature $z_u$. Furthermore, we provide the scores when the concentration of $z_u$ was 0% in training for baseline comparisons.)

## D.1   FOR CIFAR10-RESNET9: (DURING TRAINING WE HAVE **4,000** SAMPLES OF CLASS 0 IN WHICH WE INTRODUCE $z_u$ IN VARYING RATIOS (%))

Table 1: Thresholds ($\sigma_1$ and $\sigma_2$) analysis for CIFAR10-ResNet9 Setup

| Ratio (%) of $x_u$ Train Samples | Avg $MP_{4,0}$ (in %) | Avg-MP-Diff (in %) | Avg Test Accuracy (in %) | Avg-TestAcc -Diff (in %) |
|---|---|---|---|---|
| 0.0 | 9.98 | 0 | 85.28 | 0 |
| 0.03 | N/A | N/A | N/A | N/A |
| 0.05 | N/A | N/A | N/A | N/A |
| 0.1 | 8.92 | -1.05 | 83.65 | 1.63 |
| 0.15 | 8.32 | -1.65 | 86.95 | -1.67 |
| 0.2 | 8.63 | -1.35 | 85.65 | -0.37 |
| 0.25 | 10.25 | 0.28 | 85.6 | -0.32 |
| 0.3 | 10.14 | 0.17 | 84.35 | 0.93 |
| 0.35 | 10.72 | 0.74 | 83.84 | 1.44 |
| 0.4 | 12.29 | 2.32 | 84.18 | 1.1 |
| 0.45 | 11.33 | 1.35 | 83.48 | 1.8 |
| 0.5 | 12.74 | 2.76 | 84.42 | 0.86 |
| 1.0 | 18.0 | 8.02 | 82.94 | 2.34 |
| 1.5 | 24.31 | 14.33 | 79.52 | 5.76 |
| 2.0 | 51.17 | 41.2 | 73.8 | 11.48 |
| 2.5 | 61.47 | 51.49 | 63.76 | 21.52 |
| 3.75 | 88.89 | 78.91 | 43.46 | 41.82 |
| 5.0 | 93.85 | 83.87 | 37.98 | 47.3 |
| 7.5 | 97.0 | 87.02 | 25.84 | 59.44 |
| 10.0 | 98.72 | 88.75 | 25.1 | 60.18 |
| 12.5 | 99.43 | 89.46 | 17.34 | 67.94 |

From Table 1, we can clearly observe that by considering $\sigma_1$ = **3.5%** and $\sigma_2$ = **3%**, we can delineate **Feature Memorization** and **Feature Learning**. For instance, when the $x_u$ train samples ratio lies **between 0.1% and 0.5%, FM occurs**, with both Avg-MP-Diff and Avg-TestAcc-Diff being less than the thresholds. Now, as the $x_u$ samples ratio increases beyond **0.5%, up to 12.5%, the model transitions from FM to FL**, where Avg-MP-Diff and Avg-TestAcc-Diff values exceed $\sigma_1$ and $\sigma_2$. This analysis also aligns well with our definitions and the Grad-Cam analysis presented in Fig. 2c for the CIFAR10-ResNet9 results.

## D.2 FOR CIFAR100-GOOGLENET: (DURING TRAINING WE HAVE 400 SAMPLES OF CLASS 2 IN WHICH WE INTRODUCE $z_u$ IN VARYING RATIOS (%))

Table 2: Thresholds ($\sigma_1$ and $\sigma_2$) analysis for CIFAR100-GoogleNet Setup

| Ratio (%) of $x_u$ Train Samples | Avg MP$_{7,2}$ (in %) | Avg-MP-Diff (in %) | Avg Test Accuracy (in %) | Avg-TestAcc -Diff (in %) |
|---|---|---|---|---|
| 0.0 | 0.61 | 0 | 52.4 | 0 |
| 0.25 | N/A | N/A | N/A | N/A |
| 0.5 | N/A | N/A | N/A | N/A |
| 1.0 | 1.02 | 0.41 | 52.5 | -0.1 |
| 1.5 | 0.74 | 0.13 | 55.0 | -2.6 |
| 2.0 | 1.23 | 0.62 | 56.5 | -4.1 |
| 2.5 | 2.28 | 1.67 | 54.6 | -2.2 |
| 3.0 | 1.48 | 0.88 | 58.6 | -6.2 |
| 3.5 | 4.03 | 3.42 | 56.6 | -4.2 |
| 4.0 | 3.72 | 3.11 | 55.8 | -3.4 |
| 4.5 | 7.41 | 6.8 | 52.8 | -0.4 |
| 5.0 | 9.22 | 8.62 | 58.6 | -6.2 |
| 6.25 | 14.76 | 14.15 | 48.4 | 4.0 |
| 7.5 | 41.44 | 40.83 | 50.8 | 1.6 |
| 8.75 | 44.51 | 43.9 | 41.2 | 11.2 |
| 10.0 | 48.75 | 48.14 | 39.4 | 13.0 |
| 15.0 | 71.74 | 71.13 | 43.0 | 9.4 |
| 20.0 | 72.26 | 71.65 | 27.0 | 25.4 |
| 25.0 | 84.66 | 84.05 | 27.6 | 24.8 |
| 37.5 | 85.79 | 85.18 | 19.4 | 33.0 |
| 50.0 | 85.9 | 85.3 | 15.4 | 37.0 |
| 75.0 | 94.11 | 93.5 | 12.0 | 40.4 |

We again observe in Table 2, that for **CIFAR-100**, the same thresholds, $\sigma_1$ **= 3.5%** and $\sigma_2$ **= 3%**, perfectly fit to delineate **Feature Memorization (FM) and Feature Learning (FL)**. **FM** occurs when ratio of $x_u$ train samples lies between 1% and 3%, with both Avg-MP-Diff and Avg-TestAcc-Diff remaining below $\sigma_1$ and $\sigma_2$, respectively. Beyond **3%**, the model transitions from **FM to FL**, with Avg-MP-Diff and Avg-TestAcc-Diff exceeding the thresholds, as concentration of $z_u$ increases in training. This observation of FM-FL is further supported by the Grad-Cam analysis provided in Fig. 13c for the CIFAR100-GoogleNet results.

### D.3 FOR UTK-FACE-RESNET18: (DURING TRAINING WE HAVE 2,198 SAMPLES OF CLASS 3 IN WHICH WE INTRODUCE $z_u$ IN VARYING RATIOS (%))

Table 3: Thresholds ($\sigma_1$ and $\sigma_2$) analysis for UTK-Face-ResNet18 Setup

| Ratio (%) of $x_u$ Train Samples | Avg MP$_{2,3}$ (in %) | Avg-MP-Diff (in %) | Avg Test Accuracy (in %) | Avg-TestAcc -Diff (in %) |
|---|---|---|---|---|
| 0.0 | 20.68 | 0.0 | 77.38 | 0.0 |
| 0.05 | N/A | N/A | N/A | N/A |
| 0.09 | N/A | N/A | N/A | N/A |
| 0.18 | 18.82 | -1.86 | 75.55 | 1.83 |
| 0.27 | 20.45 | -0.24 | 77.33 | 0.05 |
| 0.36 | 27.86 | 7.18 | 78.49 | -1.11 |
| 0.45 | 15.75 | -4.93 | 80.79 | -3.41 |
| 0.55 | 19.77 | -0.92 | 79.85 | -2.47 |
| 0.64 | 19.79 | -0.89 | 77.18 | 0.2 |
| 0.73 | 25.41 | 3.41 | 76.8 | 0.58 |
| 0.82 | 32.36 | 6.67 | 77.58 | -0.2 |
| 0.91 | 21.3 | 0.62 | 79.99 | -2.61 |
| 1.0 | 28.51 | 7.82 | 80.93 | -3.55 |
| 1.14 | 34.98 | 14.29 | 75.25 | 2.13 |
| 1.36 | 32.16 | 11.47 | 80.17 | -2.79 |
| 1.59 | 35.54 | 14.86 | 76.22 | 1.16 |
| 1.82 | 48.64 | 27.96 | 78.81 | -1.43 |
| 2.73 | 53.57 | 32.89 | 76.83 | 0.55 |
| 4.55 | 57.92 | 37.24 | 69.61 | 7.77 |
| 9.1 | 84.59 | 63.91 | 50.33 | 27.05 |
| 13.65 | 93.75 | 73.07 | 49.58 | 27.8 |
| 18.2 | 98.79 | 78.11 | 30.66 | 46.72 |
| 22.75 | 97.68 | 76.99 | 20.93 | 56.45 |

We observe from Table 3, that the same thresholds, $\sigma_1$ = **3.5%** and $\sigma_2$ = **3%**, remain robust and perfectly applicable to distinguish **Feature Memorization (FM) and Feature Learning (FL)** for **UTK-Face-ResNet18** setup as well. **FM** occurs when the $x_u$ train samples ratio lies between **0.18%** **and 0.73%**, with both Avg-MP-Diff and Avg-TestAcc-Diff remaining below the thresholds. Beyond **0.73%**, the model transitions from **FM to FL**, as Avg-MP-Diff and Avg-TestAcc-Diff significantly exceed both $\sigma_1$ and $\sigma_2$. This observation of FM-FL is further supported by the Grad-CAM analysis provided in Fig. 9c for the UTK-Face-ResNet18 results, reinforcing our definitions of FM and FL even further.

## D.4 FOR UTK-FACE-VGG16: (DURING TRAINING WE HAVE 2,198 SAMPLES OF CLASS 3 IN WHICH WE INTRODUCE $z_u$ IN VARYING RATIOS (%))

Table 4: Thresholds ($\sigma_1$ and $\sigma_2$) analysis for UTK-Face-VGG16 Setup

| Ratio (%) of $x_u$ Train Samples | Avg MP$_{2,3}$ (in %) | Avg-MP-Diff (in %) | Avg Test Accuracy (in %) | Avg-TestAcc -Diff (in %) |
|---|---|---|---|---|
| 0.0 | 15.57 | 0.0 | 81.28 | 0.0 |
| 0.05 | N/A | N/A | N/A | N/A |
| 0.09 | N/A | N/A | N/A | N/A |
| 0.18 | 14.81 | -0.77 | 82.97 | -1.69 |
| 0.27 | 15.04 | -0.54 | 82.79 | -1.51 |
| 0.36 | 16.42 | 0.84 | 82.82 | -1.54 |
| 0.45 | 16.73 | 1.16 | 83.26 | -1.98 |
| 0.55 | 17.16 | 1.58 | 82.91 | -1.63 |
| 0.73 | 17.63 | 2.06 | 83.06 | -1.78 |
| 0.91 | 31.6 | 16.03 | 82.53 | -1.25 |
| 1.14 | 37.31 | 21.73 | 81.92 | -0.64 |
| 1.36 | 44/63 | 29.06 | 81.34 | -0.06 |
| 1.59 | 52.51 | 36.94 | 79.94 | 1.34 |
| 1.82 | 58.78 | 43.2 | 79.07 | 2.21 |
| 2.73 | 82.28 | 66.71 | 69.49 | 11.79 |
| 4.55 | 96.45 | 80.87 | 42.3 | 38.98 |
| 9.1 | 99.38 | 83.8 | 15.23 | 66.06 |
| 13.65 | 99.69 | 84.11 | 6.99 | 74.29 |
| 18.2 | 99.76 | 84.18 | 4.57 | 76.71 |
| 22.75 | 99.76 | 84.18 | 4.22 | 77.06 |

We observe from Table 4, that the same thresholds, $\sigma_1$ **= 3.5%** and $\sigma_2$ **= 3%**, remain robust and perfectly applicable to distinguish **Feature Memorization (FM) and Feature Learning (FL)** for **UTK-Face-VGG16** setup as well. **FM** occurs when the $x_u$ train samples ratio lies between **0.18% and 0.73%**, with both Avg-MP-Diff and Avg-TestAcc-Diff remaining below the thresholds. Beyond **0.73%**, the model transitions from **FM to FL**, as Avg-MP-Diff and Avg-TestAcc-Diff significantly exceed both $\sigma_1$ and $\sigma_2$. This observation of FM-FL is further supported by the Grad-CAM analysis provided in Fig. 2c for the UTK-Face-VGG16 results, reinforcing our definitions of FM and FL even further.

## D.5 FOR NICO++, MOBILEViT: (DURING TRAINING WE HAVE 720 SAMPLES OF CLASS 6 IN WHICH WE INTRODUCE $z_u$ IN VARYING RATIOS (%))

Table 5: Thresholds ($\sigma_1$ and $\sigma_2$) analysis for NICO++ - MobileViT Setup

| Ratio (%) of $x_u$ Train Samples | Avg MP$_{9,6}$ (in %) | Avg-MP-Diff (in %) | Avg Test Accuracy (in %) | Avg-TestAcc -Diff (in %) |
|---|---|---|---|---|
| 0.0 | 5.12 | 0 | 81.42 | 0 |
| 0.14 | N/A | N/A | N/A | N/A |
| 0.69 | N/A | N/A | N/A | N/A |
| 1.39 | N/A | N/A | N/A | N/A |
| 2.78 | 8.0 | 2.88 | 88.89 | -7.47 |
| 5.56 | 0.0 | -5.12 | 79.56 | 1.87 |
| 8.33 | 8.61 | 3.49 | 79.85 | 1.57 |
| 9.72 | 1.93 | -3.19 | 83.56 | -2.13 |
| 10.42 | 3.79 | -1.33 | 83.22 | -1.8 |
| 11.11 | 2.05 | -3.07 | 84.78 | -3.36 |
| 11.39 | 6.46 | 1.34 | 87.0 | -5.58 |
| 11.94 | 5.53 | 0.41 | 87.38 | -5.96 |
| 12.5 | 4.39 | -0.73 | 81.56 | -0.13 |
| 13.89 | 4.29 | -0.83 | 81.56 | -0.13 |
| 16.67 | 7.67 | 2.55 | 84.44 | -3.02 |
| 18.06 | 5.55 | 0.43 | 82.49 | -1.07 |
| 19.44 | 5.8 | 0.69 | 86.49 | -5.07 |
| 20.83 | 4.28 | -0.84 | 80.09 | 1.33 |
| 27.78 | 10.53 | 5.41 | 83.82 | -2.4 |
| 34.72 | 17.37 | 12.25 | 82.58 | -1.16 |
| 41.67 | 18.36 | 13.24 | 79.91 | 1.51 |
| 55.56 | 43.03 | 37.91 | 78.58 | 2.84 |
| 69.44 | 57.63 | 52.51 | 68.44 | 12.98 |
| 83.33 | 76.35 | 71.23 | 62.13 | 19.29 |

For **NICO++, MobileViT** as well, the thresholds, $\sigma_1$ **= 3.5% and** $\sigma_2$ **= 3%**, remain robust and perfectly applicable to distinguish **Feature Memorization (FM) and Feature Learning (FL)**, as observed in Table 5. **FM** occurs with the $x_u$ train samples ratio lying between **2.78% and 20.83%**, with both Avg-MP-Diff and Avg-TestAcc-Diff remaining below the thresholds. Beyond **20.83%**, the model transitions from **FM to FL**, as Avg-MP-Diff and Avg-TestAcc-Diff exceed the thresholds. This observation of FM-FL is further supported by the Grad-CAM analysis provided in Fig. 15c for the NICO++ results, reinforcing and validating our definitions of FM and FL.

### D.6 For Medical Abstracts-RoBERTa: (During training we have 878 samples of Class 3 in which we introduce $z_u$ in varying ratios (%))

Table 6: Thresholds ($\sigma_1$ and $\sigma_2$) analysis for Medical Abstracts-RoBERTa Setup

| Ratio (%) of $x_u$ Train Samples | Avg MP$_{0,3}$ (in %) | Avg-MP-Diff (in %) | Avg Test Accuracy (in %) | Avg-TestAcc -Diff (in %) |
|---|---|---|---|---|
| 0.0 | 10.64 | 0 | 69.17 | 0 |
| 0.11 | N/A | N/A | N/A | N/A |
| 0.23 | N/A | N/A | N/A | N/A |
| 0.46 | N/A | N/A | N/A | N/A |
| 0.68 | N/A | N/A | N/A | N/A |
| 0.91 | 9.69 | -0.95 | 68.01 | 1.16 |
| 1.14 | 9.03 | -1.61 | 69.76 | -0.59 |
| 1.37 | 13.33 | 2.69 | 76.28 | -7.11 |
| 1.59 | 13.02 | 2.38 | 68.38 | 0.79 |
| 1.82 | 12.48 | 1.84 | 67.72 | 1.45 |
| 2.05 | 11.18 | 0.54 | 73.72 | -4.55 |
| 2.28 | 6.06 | -4.58 | 73.91 | -1.89 |
| 2.51 | 11.68 | 1.04 | 67.72 | 2.52 |
| 2.96 | 11.48 | 0.84 | 68.97 | 2.52 |
| 3.42 | 16.62 | 5.98 | 67.46 | 7.76 |
| 4.56 | 58.55 | 47.91 | 54.68 | 3.14 |
| 5.69 | 45.43 | 34.79 | 42.82 | 3.35 |
| 11.39 | 94.35 | 83.71 | 10.01 | 3.35 |
| 22.78 | 100 | 89.36 | 0.0 | 10.06 |
| 34.17 | 100 | 89.36 | 0.0 | 3.77 |
| 45.56 | 100 | 89.36 | 0.0 | 5.66 |

For **Medical Abstracts-RoBERTa** as well, the thresholds, $\sigma_1$ = **3.5% and** $\sigma_2$ = **3%**, remain robust and perfectly applicable to distinguish **Feature Memorization (FM) and Feature Learning (FL)**, as observed in Table 6. **FM** occurs with the $x_u$ train samples ratio lying between **0.91% and 2.96%**, with both Avg-MP-Diff and Avg-TestAcc-Diff remaining below the thresholds. Beyond **2.96%**, the model transitions from **FM to FL**, as Avg-MP-Diff and Avg-TestAcc-Diff exceed the thresholds. This observation of FM-FL is further supported by the SHAP score analysis provided in Fig. 4c for the Medical Abstracts dataset results, reinforcing and validating our definitions of FM and FL.

### D.7 FOR EMOTIONS-BERT: (DURING TRAINING WE HAVE 572 SAMPLES OF CLASS 4 IN WHICH WE INTRODUCE $z_u$ IN VARYING RATIOS (%))

Table 7: Thresholds ($\sigma_1$ and $\sigma_2$) analysis for Emotions-BERT Setup

| Ratio (%) of $x_u$ Train Samples | Avg MP$_{2,4}$ (in %) | Avg-MP-Diff (in %) | Avg Test Accuracy (in %) | Avg-TestAcc-Diff (in %) |
|---|---|---|---|---|
| 0.0 | 0.0 | 0 | 92.45 | 0 |
| 0.17 | N/A | N/A | N/A | N/A |
| 0.35 | N/A | N/A | N/A | N/A |
| 0.7 | N/A | N/A | N/A | N/A |
| 1.05 | N/A | N/A | N/A | N/A |
| 1.4 | N/A | N/A | N/A | N/A |
| 1.75 | N/A | N/A | N/A | N/A |
| 2.1 | 0 | 0 | 90.25 | 2.2 |
| 2.45 | 0 | 0 | 91.19 | 1.26 |
| 2.8 | 0 | 0 | 92.45 | 0.0 |
| 3.5 | 0 | 0 | 93.08 | -0.63 |
| 5.24 | 0 | 0 | 92.57 | -1.89 |
| 6.99 | 6.27 | 6.27 | 89.94 | 2.52 |
| 7.87 | 3.7 | 3.7 | 89.94 | 2.52 |
| 8.74 | 17 | 17 | 84.7 | 7.76 |
| 10.49 | 18.27 | 18.27 | 89.31 | 3.14 |
| 11.36 | 15.82 | 15.82 | 89.1 | 3.35 |
| 12.24 | 23.04 | 23.04 | 89.1 | 3.35 |
| 13.99 | 28.23 | 28.23 | 82.39 | 10.06 |
| 14.86 | 31.08 | 31.08 | 88.68 | 3.77 |
| 17.48 | 35.72 | 35.72 | 86.79 | 5.66 |
| 26.22 | 70.56 | 70.56 | 68.13 | 24.32 |
| 34.97 | 76.22 | 76.22 | 44.65 | 47.8 |
| 52.45 | 98.93 | 98.93 | 21.38 | 71.07 |
| 69.93 | 97.87 | 97.87 | 4.82 | 87.63 |
| 87.41 | 100 | 100 | 1.68 | 90.78 |

For **Emotions-BERT** as well, the thresholds, $\sigma_1$ **= 3.5% and** $\sigma_2$ **= 3%**, remain robust and perfectly applicable to distinguish **Feature Memorization (FM) and Feature Learning (FL)**, as observed in Table 7. **FM** occurs with the $x_u$ train samples ratio lying between **2.1% and 5.24%**, with both Avg-MP-Diff and Avg-TestAcc-Diff remaining below the thresholds. Beyond **5.24%**, the model transitions from **FM to FL**, as Avg-MP-Diff and Avg-TestAcc-Diff exceed the thresholds. This observation of FM-FL is further supported by the SHAP score analysis provided in Fig. 17c for the Emotions dataset results, reinforcing and validating our definitions of FM and FL.

### D.8 FOR EMOTIONS-ROBERTA: (DURING TRAINING WE HAVE 572 SAMPLES OF CLASS 4 IN WHICH WE INTRODUCE $z_u$ IN VARYING RATIOS (%))

Table 8: Thresholds ($\sigma_1$ and $\sigma_2$) analysis for Emotions-RoBERTa Setup

| Ratio (%) of $x_u$ Train Samples | Avg MP$_{3,4}$ (in %) | Avg-MP-Diff (in %) | Avg Test Accuracy (in %) | Avg-TestAcc -Diff (in %) |
|---|---|---|---|---|
| 0.0 | 46.64 | 0 | 88.24 | 0 |
| 0.17 | N/A | N/A | N/A | N/A |
| 0.35 | N/A | N/A | N/A | N/A |
| 0.7 | N/A | N/A | N/A | N/A |
| 1.05 | N/A | N/A | N/A | N/A |
| 1.4 | N/A | N/A | N/A | N/A |
| 1.75 | N/A | N/A | N/A | N/A |
| 2.1 | N/A | N/A | N/A | N/A |
| 2.45 | N/A | N/A | N/A | N/A |
| 2.8 | N/A | N/A | N/A | N/A |
| 3.5 | 47.69 | 1.05 | 88 | 0.24 |
| 3.85 | 38.8 | -7.84 | 91 | -2.67 |
| 4.2 | 45.57 | -1.07 | 85.64 | 2.91 |
| 4.55 | 43.06 | -3.59 | 86.91 | 1.7 |
| 5.24 | 44.41 | -2.24 | 85.64 | 1.33 |
| 6.12 | 47.89 | 1.25 | 86.91 | 2.61 |
| 6.99 | 48.69 | 2.04 | 86.91 | 1.33 |
| 10.49 | 53.95 | 7.31 | 84.18 | 4.06 |
| 13.99 | 57.46 | 10.82 | 84.85 | 3.39 |
| 17.48 | 69.95 | 23.31 | 82.42 | 5.82 |
| 26.22 | 73.19 | 26.55 | 75.64 | 12.61 |
| 34.97 | 79.07 | 32.42 | 73.45 | 14.79 |
| 52.45 | 91.51 | 44.86 | 55.27 | 32.97 |
| 69.93 | 95.61 | 48.96 | 44.85 | 43.39 |
| 87.41 | 98.59 | 51.95 | 22.06 | 66.18 |

For **Emotions-RoBERTa** as well, the thresholds, $\sigma_1$ **= 3.5% and** $\sigma_2$ **= 3%**, remain robust and perfectly applicable to distinguish **Feature Memorization (FM) and Feature Learning (FL)**, as observed in Table 8. **FM** occurs with the $x_u$ train samples ratio lying between **3.5% and 6.99%**, with both Avg-MP-Diff and Avg-TestAcc-Diff remaining below the thresholds. Beyond **6.99%**, the model transitions from **FM to FL**, as Avg-MP-Diff and Avg-TestAcc-Diff exceed the thresholds. This observation of FM-FL is further supported by the SHAP score analysis provided in Fig. 18c for the Emotions dataset results, reinforcing and validating our definitions of FM and FL.

### D.9 CONCLUDING REMARKS

To the best of our knowledge, **no prior work has established numerical thresholds specifically for distinguishing between FM and FL, making this a significant aspect of our study.** We want to emphasize that, although these thresholds are empirical decisions, we observe consistent values for $\sigma_1$ and $\sigma_2$ across **all** datasets. These values are **nearly identical for each dataset, with $\sigma_1$ = 3.5% and $\sigma_2$ = 3%**. This consistency highlights the robustness of our approach, as we applied the same decision thresholds across all experiments. Additionally, we run experiments across **5 different random seeds for each dataset**, further strengthening the reliability of our metrics and the consistency of the thresholds.

