# OpenReview forum: "Memorization is Not Learning: Delineated through Features and Labels"
_ICLR.cc/2026/Conference — ICLR 2026 Conference Withdrawn Submission_

### Official Review · Reviewer_5io2 · 2025-10-27

**Soundness:** 2
**Presentation:** 3
**Contribution:** 2
**Rating:** 4
**Confidence:** 4

**Summary:**

This paper delineate memorization and learning at the feature level and to examine their interactions with label memorization (LM). Through controlled experiments on both vision and language datasets, the authors propose a quantitative framework involving Sensitivity Score (Ssens), Average Misclassification Difference (Avg-MP-Diff), and Average Test Accuracy Difference (Avg-TestAcc-Diff).

**Strengths:**

1. Proposed reproducible quantitative metrics (Ssens, Avg-MP-Diff)
2. Cross-modal experiments show consistent trends across both vision and text domains

**Weaknesses:**

1. Defination of Feature memorization and Feature learning overlaps with traditional memorization/generalization; limited novelty.
2. The finding that noisy labels induce spurious feature learning strongly similar to conclusion from Feldman[1], but the paper fails to clearly delineate how its contribution extends beyond prior work.
3. Artificially injected features may bring extra distribution shift or work similar as noise. How would the quality or semantic alignment of the features will influence the results is not discussed.

[1] Feldman, Vitaly. "Does learning require memorization? a short tale about a long tail." Proceedings of the 52nd annual ACM SIGACT symposium on theory of computing. 2020.

**Questions:**

1. How do Feature learning different from the "generalization" in previous works and Feature memorization differ from the memorization?
2. Is all Feature Learning necessarily desirable, especially when induced by noisy labels? Or is all Feature memorization harm the model? Some reasearches in SSL-Vision domain also point out that the memorization of outliers could contributes to model's generlization.[1]
3. How sensitive are FM and FL to model architecture? Is feature memorization influenced by architectural properties such as locality or inter-layer dependencies?



[1] Wang, Wenhao, et al. "Memorization in Self-Supervised Learning Improves Downstream Generalization." The Twelfth International Conference on Learning Representations.

---

### Official Review · Reviewer_ndtS · 2025-10-30

**Soundness:** 2
**Presentation:** 2
**Contribution:** 2
**Rating:** 2
**Confidence:** 5

**Summary:**

The paper studies feature-level memorization and feature learning, in contrast to prior literature that focuses on label-level memorization. It introduces a framework and definitions to study these phenomena. Through experiments across both vision and language models, the authors find that label memorization suppresses feature memorization while inducing feature learning.

**Strengths:**

This work provides a clear and formal distinction between feature memorization and feature learning, addressing a gap in prior research that mostly examined memorization at the sample level. It introduces quantitative metrics and thresholds to systematically identify when a model recognizes a feature versus when it generalizes it. The comprehensive experiments across multiple modalities and architectures strengthen the generality and robustness of the findings.

**Weaknesses:**

**Definition Not Based on Self-Influence:**

The paper defines memorization in terms of generalization to the test set, rather than self-influence, which is how memorization is commonly studied in the literature i.e., by measuring changes in accuracy on a specific point when it is excluded from training. This shift introduces key limitations: 1. The test set distribution is independent of individual training points and can vary across datasets. 2. Test set leakage (duplicate samples between train and test splits) further complicates such evaluations. 3. Departing from self-influence marks a fundamental conceptual shift, as the focus moves from the behavior of the point itself to that of unseen data.

While alternative definitions are welcome, the authors should 1) explicitly acknowledge that their approach diverges from convention and 2) clearly justify why this new framing is necessary.

**Does Not Account for Ground Truth:**

The definition also fails to consider how applying the feature Zu changes the final results if the points are naturally memorized (e.g., by Feldmen et al's definition). The authors’ own results show that mislabeled points (artificial memorization) behave differently when Zu​ features are introduced. This raises a natural question: How does Zu impact naturally memorized points, which is the basis of their experiments. Therefore, before applying Zu transformations, they should confirm whether the target points are subject to label memorization in the first place.


**Motivation:**

The motivation for the study is underdeveloped. The authors should clarify why understanding feature-level memorization is important for the broader memorization literature. Currently, the paper reads as a collection of disconnected experiments rather than a cohesive argument. The main result that, few samples with Zu​ features are memorized while more samples lead to generalization, is not particularly novel, as it parallels the established finding that outlier samples tend to be memorized. The contribution would be strengthened by emphasizing how label memorization can facilitate feature learning, thereby connecting the findings to a broader conceptual insight.


**Writing and Reproducibility:**
The writing requires significant improvement for clarity and coherence. For example, the abstract begins with a circular definition for memorization, which can confuse readers. In addition, the paper lacks sufficient detail on how “in” and “out” sets are constructed. While prior work (e.g., Feldman et al.) specifies dropping 30% of the data, no such information is provided here. This omission limits the interpretability of the sensitivity-based memorization metric.

**Questions:**

Find above

---

### Official Review · Reviewer_vcDu · 2025-10-30

**Soundness:** 2
**Presentation:** 2
**Contribution:** 2
**Rating:** 2
**Confidence:** 3

**Summary:**

The paper proposes a clear operational distinction between Feature Memorization (FM) and Feature Learning (FL), introduces metrics to detect each, and studies how Label Memorization (LM) interacts with them. A sensitivity score S_{\text{sens}} tests whether a model has recognized a tracked feature z_u, and two deltas—Avg-MP-Diff (misclassification shift) and Avg-TestAcc-Diff (accuracy shift)—separate FM (recognized but non-generalizing) from FL (recognized and generalizing). Thresholds \sigma_1=3.5\% and \sigma_2=3\% determine when shifts are “significant.” Across image and text classifiers, the authors report three regimes as the proportion of z_u in training grows: no recognition \rightarrow FM \rightarrow FL. They further claim that when z_u co-occurs with noisy labels, LM suppresses FM and induces FL, often from very small proportions, and that this coupling emerges early in training. Evidence comes from multiple datasets (CIFAR-10/100, UTK-Face, NICO++, Emotions, Medical Abstracts) and models (CNNs, MobileViT, BERT/RoBERTa), plus Grad-CAM/SHAP analyses.

Key empirical illustrations include the three-zone transition on UTK-Face and Medical Abstracts (e.g., Fig. 2 on p.5 and Fig. 4 on p.6) and LM-driven FL with fixed noisy labels (e.g., Figs. 6–8, pp.8–9). The training-time analyses show S_{\text{sens}} and LM score rising in tandem as training proceeds (e.g., Figs. 3 & 7).

**Strengths:**

•	Originality: Clear, operational definitions for FM vs. FL with concrete decision criteria; systematic exploration of how LM reshapes feature behavior (suppression of FM, induction of FL).
•	Quality: Multi-dataset, multi-model study; per-epoch analyses that align S_{\text{sens}} and LM-score trajectories with performance gaps; consistent three-zone pattern.
•	Clarity: Definitions, metrics, and evaluation procedures are spelled out, including a one-tailed t-test for recognition.
•	Significance: Highlights a concrete mechanism by which label noise can cause harmful feature generalization—actionable for robust training and dataset curation.

**Weaknesses:**

1.	Threshold sensitivity & statistics. The global thresholds (\sigma_1,\sigma_2) are fixed across datasets without principled calibration; there’s no sensitivity analysis, multiple-testing control, or confidence intervals on Avg-MP-Diff / Avg-TestAcc-Diff. This risks categorization instability near decision boundaries.
2.	Synthetic features & external validity. Overlaid strings (“JOHN DOE”, “Riverdale”, letters) are convenient but may not reflect real spurious cues (textures, backgrounds, co-occurring objects). It’s unclear whether the same FM→FL transitions and LM effects hold on established spurious-correlation benchmarks without feature injection.
3.	Training regime amplifies memorization. Forcing ~100% train accuracy and disabling data augmentation likely magnify FM/FL effects; the paper doesn’t probe whether standard regularization/augmentation breaks or shifts the reported regimes.
4.	Interpretability reliance. Conclusions often lean on Grad-CAM/SHAP—for which faithfulness can vary—without alternate causal probes (counterfactual edits beyond z_u, representational tests). Evidence would be stronger with feature-ablation or causal mediation analyses. (Figures across pp.5–9.)
5.	Limited theoretical framing. The work posits empirical boundaries (low proportion→FM; high→FL) but offers no theory explaining why the transition thresholds arise or how they depend on dataset/architecture scale.

**Questions:**

1.	Threshold robustness: How sensitive are FM/FL decisions to \sigma_1,\sigma_2? Could you report ROC-style analyses or bootstrap CIs for Avg-MP-Diff and Avg-TestAcc-Diff across seeds?
2.	Out-of-the-box training: Do your findings persist under common regularizers—MixUp/CutMix, RandAugment, weight decay sweeps, early-learning regularization, or label-smoothing?
3.	Real spurious cues: Can you replicate the FM→FL and LM-induced FL on datasets with native spurious correlations (e.g., background/texture biases) without inserting z_u? What are the observed transition ratios there?
4.	Causal tests: Beyond Grad-CAM/SHAP, can you provide counterfactual or causal-feature tests (e.g., feature scrubbing, representation linear probes) to confirm that learned associations truly mediate predictions in FL regimes?
5.	Theory/estimation: Is there a simple model predicting the FM→FL transition as a function of class size, model capacity, and feature frequency? How does label noise rate interact with that boundary?

---

### Official Review · Reviewer_ZPZy · 2025-10-31

**Soundness:** 3
**Presentation:** 2
**Contribution:** 2
**Rating:** 4
**Confidence:** 4

**Summary:**

This paper dives into the dynamics of memorization at feature level. In particular, it defines and distinguishes feature memorization vs. feature learning. By introducing a controlled feature into datasets, the authors are able to design experiments that checks how feature memorization/learning and label memorization interact with each other along the horizon of training epochs. The empirical results suggest that label memorization can suppress feature memorization and let feature learning kick in at an earlier time.

**Strengths:**

The methodology of the paper is interesting. Although the idea of introducing a controlled feature stems from previous work, the experiment design is solid. The notion of feature memorization vs. feature learning is interesting too. Overall, the empirical evaluation reveals some insights into an interesting topic.

**Weaknesses:**

My main concerns are over 1) the presentation of the paper and 2) the implication/impact of the results.

**Presentation**

The writing of the paper is below the bar of a typical ICLR paper. Many sentences are heavily comma separated or running with too many "and". It took me way longer time than average to understand the claims of the paper. The authors also uses bold font in many places to highlight the key sentence. However, these highlighted sentences themselves are not very clear/succinct. The better way to really highlight the key points is more succinct/concise writing.

**Impact**

I appreciate the thorough discourse of feature memorization/learning and their interaction with the so called label memorization. However, I don't see how the findings in this work can guide us in real model training. These concepts are studied in a controlled experiment with artificially introduced feature. However, it is unclear what a practitioner should leverage these findings in real-world tasks. See my questions in the next question.

**Questions:**

1) Could you give us more insights on why label memorization suppress feature memorization? It occurs to me that the artificial noise in label is only associated with the artificially introduced feature. In order to fit those noisy labels, the model has to learn the feature. The gain in feature learning is a consequence of the accident high correlation between noisy labels and the introduced feature.

2) Can you think of an experiment to prove/disprove my hypothesis in 1)

3) If label memorization promotes feature learning, does that mean one can introduce artificial noisy label in training to enhance learning? This sounds a bit awkward. The missing link, I believe, is how enhanced feature learning translates to better/worse model performance.

---

### Note · Authors · 2025-11-26

I have read and agree with the venue's withdrawal policy on behalf of myself and my co-authors.